# Phosphorylation-dependent tuning of mRNA deadenylation rates

James A. W. Stowell[1], Conny W. H. Yu[1], Zhuo A. Chen [2], Lily K. DeBell[1], Giselle C. Lee[1], Tomos Morgan[1], Ludwig Sinn [2], Sylvie Agnello [1], Francis J. O'Reilly [2], Juri Rappsilber [2,3], Stefan M. V. Freund [1] & Lori A. Passmore [1]✉

Shortening of messenger RNA poly(A) tails by the Ccr4–Not complex initiates mRNA decay and is a major determinant of gene regulation. RNA adaptors modulate the specificity of deadenylation by binding to Ccr4–Not through their intrinsically disordered regions (IDRs). However, the determinants of specificity and their regulation are largely unclear. Here we use nuclear magnetic resonance spectroscopy, biochemical reconstitution and structural modeling to show that dispersed segments within the IDR of the fission yeast Puf3 RNA adaptor interact with Ccr4–Not, consistent with multivalency. Binding can be modulated by phosphorylation, altering the deadenylation rate in a continuously tunable manner. Regulation of deadenylation through multivalency and phosphorylation likely occurs in evolutionarily divergent IDRs from additional RNA adaptors, including human Pumilio and Tristetraprolin. Overall, our in vitro data suggest that mRNA decay can be regulated not only as a bistable on–off switch but also by a graded mechanism, rationalizing how post-transcriptional gene expression can be fine-tuned.

Regulation of mRNA 3′-poly(A) tail length is critical for the control of mRNA half-life and translational efficiency[1]. Shortening of poly(A) tails is mediated by Ccr4–Not and Pan2–Pan3, highly conserved multiprotein complexes that contain exonucleases termed deadenylases[2–5]. Ccr4–Not contains seven core subunits including the deadenylase enzymes Ccr4 (CNOT6 and CNOT6L; subunit names are listed in uppercase for humans) and Caf1 (CNOT7 and CNOT8), which assemble around the ~200-kDa Not1 (CNOT1) scaffold[3] (Extended Data Fig. 1a).

Ccr4–Not is targeted to specific mRNAs through a diverse range of RNA adaptor proteins that respond to cellular signals, developmental cues and the translational state of the ribosome[1]. RNA adaptor proteins commonly have a modular architecture, including a structurally conserved RNA-binding domain that interacts with specific *cis*-acting RNA sequence elements and an intrinsically disordered region (IDR) that recruits Ccr4–Not. RNA adaptors, therefore, tether specific transcripts to the deadenylation machinery to accelerate poly(A) tail shortening[1,6]. In this 'tethering model', short linear motifs (SLiMs) bind to one of several hydrophobic pockets located on the Not1, Not3 (CNOT3) and Not9 (also known as Caf40 or Rcd1 in yeast; CNOT9 in humans) subunits (Extended Data Fig. 1a). For example, distinct SLiMs in the RNA-binding proteins UnKempt and Roquin interact with both CNOT9 and the C-terminal NOT module (a structured region of Not1, Not2 and Not3)[7,8].

Disruption of SLiMs in RNA adaptors reduces their ability to bind Ccr4–Not, thereby decreasing RNA turnover. However, a single SLiM is often insufficient to account for the full repressive activity of the IDR, both in vitro and in vivo. For example, the AU-rich element (ARE) binding protein Tristetraprolin (TTP or ZFP36) interacts with an N-terminal HEAT repeat in CNOT1 through a highly conserved RLP(φ)F SLiM (where φ is a hydrophobic residue)[9]. Deletion of this motif only partially stabilizes proinflammatory TTP-targeted mRNAs in vivo and does not have the severe autoimmune phenotype of knockout mice[10]. Thus, it is likely that additional elements within TTP contribute to its binding to Ccr4–Not and stimulation of deadenylation.

Discrete SLiMs that are necessary and sufficient for repressive activity have been difficult to identify in some RNA adaptors.

[1]MRC Laboratory of Molecular Biology, Cambridge, UK. [2]Chair of Bioanalytics, Technische Universität Berlin, Berlin, Germany. [3]Wellcome Centre for Cell Biology, University of Edinburgh, Edinburgh, UK. ✉e-mail: passmore@mrc-lmb.cam.ac.uk

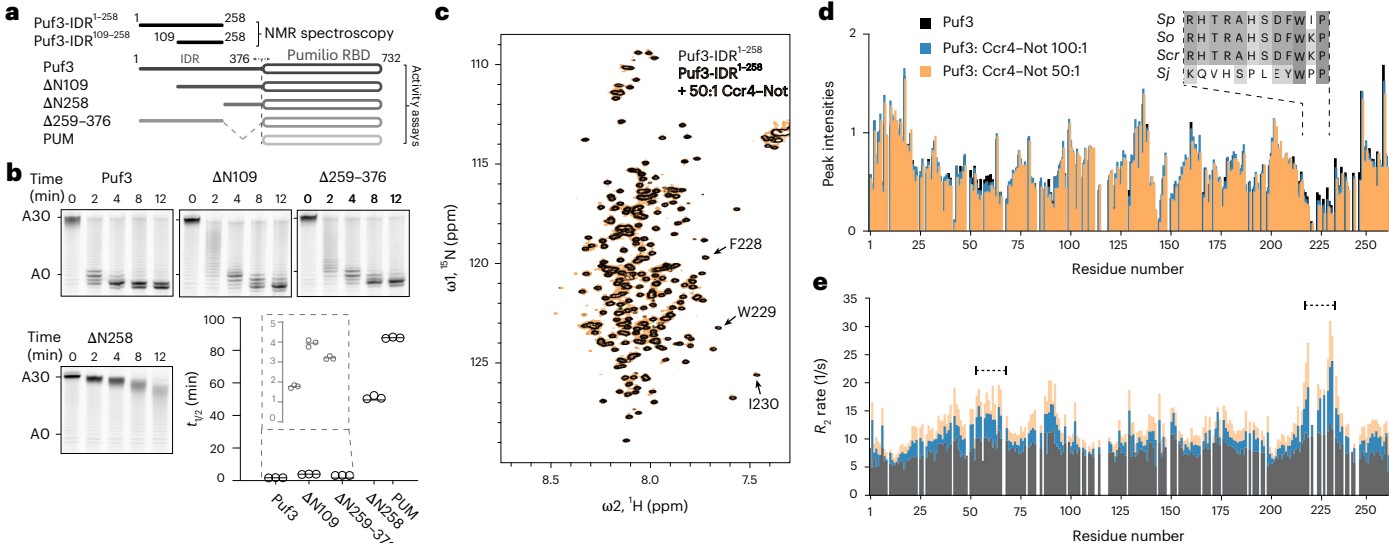

**Fig. 1 | An N-terminal IDR of Puf3 is required for stimulation of deadenylation by Ccr4–Not. a**, Domain diagrams of *S. pombe* Puf3 constructs used in this study. Puf3 contains a 376-aa N-terminal IDR and a C-terminal Pumilio RNA-binding domain (RBD) or PUM. Puf3-IDR[1–258] and Puf3-IDR[109–258] NMR constructs are indicated above the full-length protein. Truncation constructs used for deadenylation assays in **b** are indicated. **b**, Deadenylation assays using recombinant *S. pombe* Ccr4–Not complex, MBP-tagged full-length or truncated Puf3 constructs and 5′ fluorescently labeled substrate RNA containing a Pumilio recognition element upstream of a 30-nt poly(A) tail. The reaction mixture was resolved using denaturing PAGE at the indicated time points. The sizes of substrate RNA with a 30-nt poly(A) tail (A30) or no poly(A) tail (A0) are indicated. Deadenylation rates are plotted as $t_{1/2}$. Experiments were performed in triplicate (open circles) and averaged (line). **c**, ¹H,¹⁵N 2D-HSQC of 50 μM Puf3-IDR[1–258]

with (orange) or without (gray) 1 μM Ccr4–Not. Addition of Ccr4–Not at this concentration resulted in minimal exchange broadening. Crosspeaks of the DFW motif are indicated. **d,e**, Peak intensities (**d**) and transverse relaxation rates ($R_2$) (**e**) of Puf3-IDR[1–258] from ¹H,¹⁵N 2D-HSQC experiments alone (black) and in the presence of Ccr4–Not at 50:1 (orange) and 100:1 (blue) ratios. The increase in $R_2$ across the whole IDR is consistent with the conformational ensemble sampling bound states. Higher $R_2$ rates in several distinct regions are indicative of additional contributions, from both direct interactions with Ccr4–Not and the exchange regime between free and bound states. Dashed lines indicate two regions with changes in $R_2$ that were mutated in Fig. 2a. Conservation of the DFW motif in Schizosaccharomycetes is shown in the sequence alignment in **d**. *Sp*, *S. pombe*; *So*, *Schizosaccharomyces octosporus*; *Scr*, *Schizosaccharomyces cryophilus*; *Sj*, *Schizosaccharomyces japonicus*.

For example, *Drosophila* Pumilio and human PUM1 contain long IDRs with multiple regions that can repress reporter mRNAs through interaction with Ccr4–Not[11–14]. The large size, low sequence conservation and lack of structure in these IDRs generate challenges in studying their binding mechanisms. Thus, despite multiple lines of evidence supporting the essentiality of IDRs and SLiMs in RNA turnover, mechanistic details are lacking.

Here, we use in vitro reconstitution and structural biology to address how RNA adaptors bind and stimulate Ccr4–Not. We find that they act through a multipartite IDR-binding mechanism that is tuned by phosphorylation. This is reminiscent of a mechanism used by transcription factors to regulate mRNA synthesis[15,16], suggesting commonalities across the regulation of gene expression.

## Results

### Multiple regions in the Puf3-IDR are required for stimulation of Ccr4–Not

To understand how the specificity of mRNA deadenylation is mediated, we first investigated how the *Schizosaccharomyces pombe* Pumilio RNA adaptor Puf3 binds Ccr4–Not. Puf3 promotes targeted deadenylation in a reconstituted in vitro assay with Ccr4–Not and stimulates deadenylation ~50-fold[17] (Fig. 1a,b and Extended Data Fig. 1b–d). N-terminal and C-terminal truncations of the N-terminal IDR in Puf3 lead to progressive decreases in activity and the first 258 residues of the IDR are necessary for efficient stimulation of deadenylation (Fig. 1b).

Next, we used solution-state nuclear magnetic resonance (NMR) spectroscopy to gain structural insights into the Puf3-IDR. We assigned the backbone resonances of the 258-residue Puf3-IDR, hereafter referred to as Puf3-IDR[1–258], using a combination of ¹H-detected and ¹³C-detected experiments, allowing us to obtain a near-complete assignment (249 of 258 backbone amide and imino resonances) (Fig. 1c

and Extended Data Fig. 2a,b). We also used a shorter construct spanning residues 109–258 (Puf3-IDR[109–258]) to confirm the obtained assignments (Extended Data Fig. 2b). Overall, the narrow dispersion of ¹H chemical shifts in the ¹H,¹⁵N two-dimensional (2D) heteronuclear single quantum correlation (HSQC) spectrum and modest secondary chemical shifts confirm that Puf3-IDR[1–258] is largely disordered in solution (Fig. 1c and Extended Data Fig. 2c).

To characterize the interaction with Ccr4–Not, we titrated the full-length 0.5-MDa unlabeled complex (containing Not1, Not2, Not3, Not4, Not9, Ccr4 and Caf1) into isotopically labeled Puf3-IDR[1–258] (Extended Data Fig. 2d,e). Concentration-dependent peak attenuation was observed in the spectra, indicating that the Puf3-IDR[1–258] interacts with Ccr4–Not. High concentrations of Ccr4–Not resulted in a complete loss of signal because of the high molecular weight of the bound complex. Thus, we recorded data with lower concentrations that minimally perturbed the spectra (Fig. 1c). The mapped peak intensities in these spectra show modest changes in several regions (Fig. 1d).

We also measured ¹⁵N transverse relaxation rates ($R_2$) of Puf3-IDR[1–258] in the presence of Ccr4–Not, which can give an indication of which regions of Puf3 directly interact (Fig. 1e). We observed a small increase in $R_2$ across the whole IDR and more substantial increases in several distinct regions such as around residues 40–90 and 227–231 (Fig. 1e). This is consistent with a direct and dynamic interaction with Ccr4–Not, where the IDR is sampling both bound and unbound states. Previously characterized SLiMs that bind Ccr4–Not contain aromatic and hydrophobic residues[18]. For example, tryptophan-containing motifs in GW182 proteins are important for interaction with CNOT1 (refs. 19,20). Notably, a conserved tryptophan-containing motif in Puf3 (residues 227–231, DFWIP in *S. pombe*, henceforth referred to as the DFW motif; Supplementary Fig. 1) shows signal attenuation and substantially increased $R_2$ in the presence of Ccr4–Not (Fig. 1d,e).

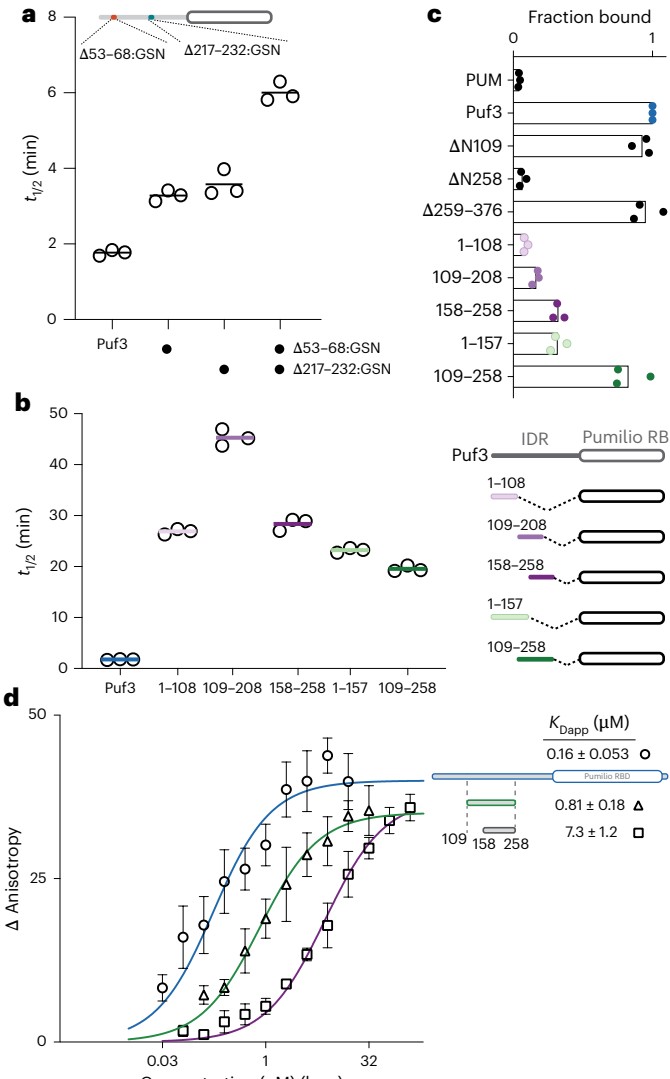

**Fig. 2 | Multiple regions of the Puf3-IDR interact with Ccr4–Not. a,b,** Relative deadenylation rates of Ccr4–Not with mutant Puf3 proteins, plotted as $t_{1/2}$. Experiments were performed in triplicate (open circles) and averaged (line). In **a**, Puf3 residues 53–68 and/or 217–232 were substituted with a GSN linker. In **b**, the Puf3-IDR was truncated and the indicated sequences were fused to the PUM RBD. **c,** Pulldown assays of truncated proteins used in **b** and Fig. 1b. MBP-tagged proteins were immobilized on amylose resin and tested for their ability to pull down Ccr4–Not. Fractional binding was quantified relative to the full-length protein. Bars denote the mean of three triplicate experiments (filled circles). **d,** Fluorescence anisotropy equilibrium analysis of Puf3 binding to labeled Ccr4–Not. Increasing concentrations of the indicated Puf3 constructs (109–258 and 158–258 are GST–lipoyl tag fusions with no PUM domain) were titrated into 25 nM *S. pombe* Ccr4–Not containing an FITC label on the Not9 subunit. Binding isotherms were fitted with a quadratic binding equation. Symbols represent the mean of three replicates with error bars showing the s.d. Calculated apparent $K_D$ values (±s.d.) are shown.

To test the functional role of the Puf3 DFW motif, we used targeted deadenylation assays with Ccr4–Not. A purified Puf3 variant in which residues 217–232 were replaced with a GSN linker had ~2-fold reduced ability to stimulate Ccr4–Not-mediated deadenylation compared to wild-type Puf3 (Fig. 2a and Extended Data Fig. 3a). Residues 53–68 of Puf3 include three phenylalanines and also showed signal attenuation and increased $R_2$ rates in the presence of Ccr4–Not. Replacement of residues 53–68 with a GSN linker also resulted in a ~2-fold reduction in activity. Combining both substitutions led to a further reduction,

consistent with a role for multiple regions in the interaction with Ccr4–Not. Individual substitutions of hydrophobic residues, either alone or in combination, also resulted in modest changes in the ability of Puf3 to stimulate Ccr4–Not (Extended Data Fig. 3b–d).

Deletion of the entire N-terminal IDR (Puf3$^{\Delta N258}$) reduces deadenylation to a much greater extent than any of our more targeted mutants (Fig. 1b). This suggests that no single region within the IDR makes a dominant contribution to the interaction with Ccr4–Not. Thus, we made further truncation variants of Puf3, which confirmed that most tested segments of Puf3-IDR$^{1–258}$ contribute to activating deadenylation and deletions of longer segments of the IDR have a more substantial effect on deadenylation than any of the amino acid substitutions (Fig. 2b and Extended Data Fig. 4a–c).

To test whether differences in activity correlate with binding, we performed pulldown experiments. We found that shorter IDR constructs generally pull down less Ccr4–Not than longer constructs (Fig. 2c and Extended Data Fig. 4d). Notably, a construct spanning amino acids 109–258 shows higher fractional binding than 109–208 or 158–258. Overall, these data support a model where extensive regions of the Puf3-IDR contribute to the interaction with Ccr4–Not.

To quantitate the roles of different regions of Puf3 on binding to Ccr4–Not, we prepared a fluorescently labeled Ccr4–Not complex and measured the apparent dissociation constants ($K_{Dapp}$) for Puf3-IDR truncation constructs using fluorescence anisotropy. A construct spanning residues 158–258 including the DFW motif binds to Ccr4–Not with a $K_{Dapp}$ of ~7 μM, whereas a construct spanning residues 109–208, excluding this motif, binds with substantially lower affinity ($K_{Dapp}$ > 50 μM) (Fig. 2d and Extended Data Fig. 4e). A construct including both regions (residues 109–258) has a $K_{Dapp}$ of 0.8 μM, closer to the $K_{Dapp}$ of the full-length protein (0.1 μM) (Fig. 2d). Interestingly, substitution of residues 53–68 or 217–232, which had large increases in $R_2$, with the GSN linker did not have a major effect on binding but substitution of both regions together reduced the binding affinity to ~10 μM (Extended Data Fig. 3f). Together, these binding data, along with NMR and mutational analyses, are consistent with multiple regions across the Puf3-IDR contributing to the interaction with Ccr4–Not to stimulate deadenylation activity with no individual region being necessary or sufficient.

## Puf3 stimulates deadenylation through multiple conserved sites on Ccr4–Not

To gain insight into which surfaces of Ccr4–Not interact with Puf3, we performed crosslinking mass spectrometry (CLMS) analysis of Ccr4–Not alone and bound to Puf3 (Fig. 3a and Extended Data Fig. 5a,b). Many of the crosslinks between Puf3 and Ccr4–Not map to the Not9 subunit but there are also crosslinks to other regions including the NOT module. Not9 is known to interact with several other RNA adaptor protein IDRs through two general mechanisms: a peptide-binding groove and two tryptophan-binding pockets[7,19,20]. Most of the crosslinks between Puf3 and Not9 in our CLMS data locate in two clusters near the Not9 peptide-binding groove (Extended Data Fig. 5c,d). Thus, we hypothesized that the central part of the Puf3-IDR is in close proximity to Not9 and may bind directly to the peptide-binding groove.

To analyze binding to Ccr4–Not subunits, we developed an Alpha-Fold2-based interaction screen using 50-residue tiles of the Puf3-IDR (across residues 1–129 and 160–291; Fig. 3b and Extended Data Fig. 5e–g). We first analyzed binding to the Not9 module (Not9 with residues 1098–1318 of Not1). The highest per-residue predicted local distance difference test (pLDDT) scores were for the DFW motif interacting with the tryptophan-binding pockets of Not9. Aromatic residues in other tiles were often inserted into the tryptophan-binding pockets. This agrees with previous data suggesting that tryptophans are important for interaction with Not9 (refs. 19–21). Multiple regions of the Puf3-IDR were also placed into the peptide-binding groove of Not9 (Fig. 3b,c and Extended Data Fig. 5e). We, therefore, hypothesize that Puf3 binds

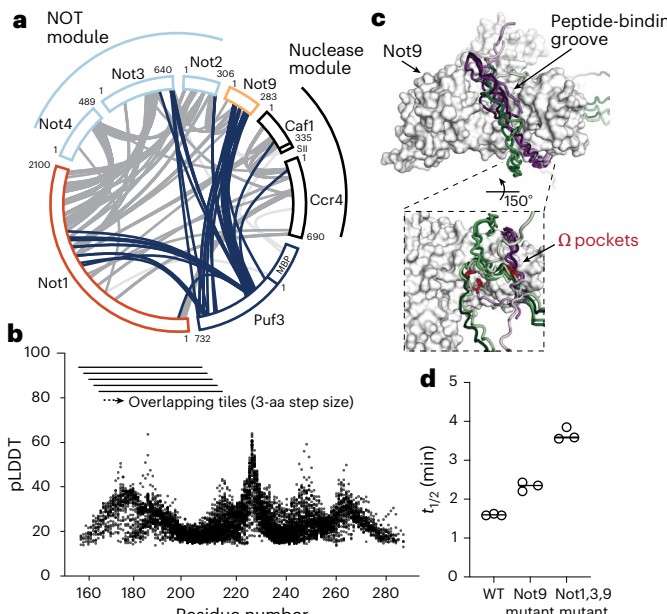

**Fig. 3 | Puf3 interacts with multiple binding sites on Ccr4–Not. a**, Sulfo-SDA-based CLMS analysis of Ccr4–Not bound to Puf3. Crosslinks highlighted in blue show interprotein crosslinks between Puf3 and Ccr4–Not subunits. Other intersubunit crosslinks are shown in gray. Subunits are colored and arranged according to the subcomplex architecture of Ccr4–Not. The N-terminal and C-terminal residue numbers for each subunit are shown. **b**, AlphaFold2 screen of 50-residue fragments of Puf3, tiled in three-residue increments, with the Not9 module (Not9 plus Not1 residues 1099–1318). The pLDDT score per residue is plotted. **c**, AlphaFold2 structure prediction showing overlay of the ten highest-scoring Puf3 fragments based on pLDDT scores (purple and green) on Not9–Not1 (white surface). The Puf3 fragments are predicted to bind in the peptide-binding groove and tryptophan (aromatic, Ω) pockets of Not9. Hydrophobic residues that insert into the Ω pockets in the predicted structures are shown as dark-red sticks. **d**, Deadenylation activities (plotted as $t_{1/2}$) of wild-type (WT) Ccr4–Not and complexes containing mutations impacting IDR-binding pockets with WT Puf3. Bars denote the mean of three independent assays (filled circles). The mutants (listed in Extended Data Fig. 6a) do not substantially affect the activity of Ccr4–Not without Puf3 (Extended Data Fig. 6c).

Not9 through both the peptide-binding groove and aromatic pockets. However, AlphaFold2 alone cannot predict with high confidence which regions within Puf3 interact with Not9.

We examined the amino acid characteristics and evolutionary conservation of the peptide-binding groove and tryptophan-binding (aromatic) pockets in Not9 and found that many interacting residues are invariant throughout eukaryotes (Extended Data Fig. 5e). To test the interaction between Puf3 and Not9, we designed substitutions of surface residues that would disrupt the conserved interaction sites (Extended Data Fig. 5e), incorporated them into Ccr4–Not and assayed the ability of Puf3 to stimulate deadenylation activity. Mutations impacting the Not9-binding sites reduced the ability of Ccr4–Not to accelerate deadenylation (~1.5–2-fold reduction) (Fig. 3d and Extended Data Fig. 6). Nevertheless, this does not account for the full repressive activity of the N-terminal IDR of Puf3 (Fig. 1b) suggesting that other binding sites on Ccr4–Not are also important for maximal acceleration of deadenylation by Puf3.

We extended our AlphaFold2-based screen to include other known IDR-binding sites on Ccr4–Not including a segment of the N-terminal HEAT repeats of Not1 (residues 640–821, the TTP-binding domain), and a C-terminal pocket in Not1 with a binding pocket on Not3 (the NOT module; Not1 residues 1568–2100, Not2 residues 74–306 and Not3 residues 471–640) (Extended Data Fig. 1a). This

suggested that the Puf3-IDR may also interact with evolutionarily conserved pockets on both Not1 and Not3 (Extended Data Fig. 5f,g). Similar to the Not9-binding site, there are a number of predicted interacting peptides and we could not identify a single binding mode of Puf3 with confidence. We introduced amino acid substitutions to disrupt the binding pockets on Not1 and Not3 and combined them with our Not9 mutants in the fully assembled recombinant mutant Ccr4–Not complex. The Not1 and Not3 mutants further reduce the ability of Puf3 to stimulate deadenylation (Fig. 3d and Extended Data Fig. 6), consistent with a contribution of these residues toward interaction with Puf3.

In summary, AlphaFold2 does not predict a single Puf3 interaction mode for each binding pocket on Ccr4–Not and our NMR, CLMS and activity data support a mechanism of interaction where multiple regions across the Puf3-IDR bind to multiple sites on the Ccr4–Not complex. These combined interactions are required for the full stimulatory activity of Puf3.

## Sequential phosphorylation of Puf3 tunes its ability to stimulate Ccr4–Not

IDRs are often substrates for post-translational modifications (PTMs) and these can be critical for their cellular function[22]. Puf3 is highly enriched in serine and threonine residues (~30% of all IDR residues) and some of these are phosphorylated in vivo[23] (Extended Data Fig. 7a). In *Saccharomyces cerevisiae*, Puf3p is phosphorylated in a glucose-responsive manner by multiple kinases including the AGC-family kinase PKA and Sch9p (ref. 24). This stabilizes Puf3p target transcripts but does not affect RNA binding by Puf3p.

To test whether the *S. pombe* Puf3-IDR is similarly phosphorylated, we incubated isotopically labeled Puf3-IDR$^{1–258}$ with the purified AGC-family kinase Sck1 and the Sck1-activating kinase Pdk1. We then separated Puf3 from the kinases and analyzed the resulting purified sample by NMR. A subset of backbone amide resonances appeared downfield shifted between 8.6 and 8.8 ppm, a region characteristic for serine and threonine phosphorylation[25] (Fig. 4a). We assigned five peaks with comparable signal intensities to the remainder of the spectrum as phosphoserine (pS42, pS156, pS207, pS226 and pS241) (Extended Data Fig. 7b). However, we also noted that there were additional peaks with lower-intensity signals that likely represent substoichiometric phosphorylations.

Time-resolved NMR experiments with Puf3-IDR$^{1–258}$ (Extended Data Fig. 7c) and intact MS (Fig. 4b,c) with Puf3-IDR$^{1–376}$ showed that Puf3 is sequentially phosphorylated by Sck1 and Pdk1 kinases. Specifically, MS showed that the full IDR was phosphorylated at 2–4 sites in the first 5 min and at up to 18 sites over the next 4.5 h (Fig. 4c). Thus, Puf3 is differentially phosphorylated under these conditions, with an increasing number of sites incorporated over time, suggesting that, in the absence of other factors, some Puf3 phosphorylation sites are preferred over others.

Interestingly, in the $^1$H,$^{15}$N 2D-HSQC spectrum, the side-chain signal of W229 (in the DFW motif) changes during progressive phosphorylation of the Puf3-IDR$^{1–258}$ (Fig. 4d). We, therefore, hypothesized that Puf3 phosphorylation may alter binding sites for Ccr4–Not (including the DFW motif) and set out to test this by solution NMR. We used the shorter Puf3-IDR$^{109–258}$ construct for NMR analysis to reduce the complexity of substoichiometric phosphorylation sites. Puf3-IDR$^{109–258}$ is phosphorylated at the same residues as Puf3-IDR$^{1–258}$ (except pS42, which is not present in this construct) (Extended Data Fig. 7d,e), confirming the selectivity of the kinases. We titrated full-length Ccr4–Not into phosphorylated Puf3-IDR$^{109–258}$ and observed less signal attenuation compared to unmodified Puf3-IDR$^{109–258}$ (Extended Data Fig. 7f). Thus, phosphorylation likely reduces Ccr4–Not interaction with the Puf3-IDR.

To further understand the effect of phosphorylation on Puf3-IDR, we investigated the backbone dynamics of the IDR. Upon

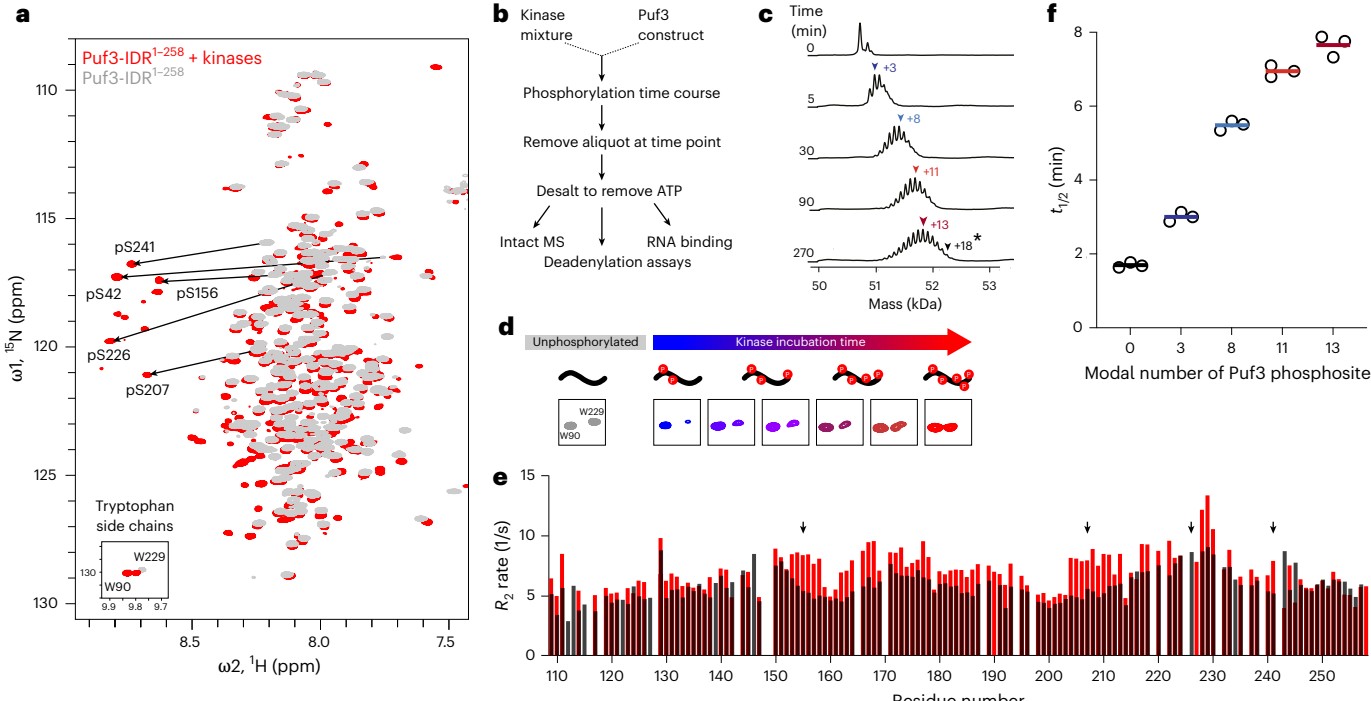

**Fig. 4 | Phosphorylation of the Puf3-IDR tunes Ccr4–Not-mediated deadenylation. a**, $^1$H,$^{15}$N 2D-HSQC spectral overlay of 75 μM unphosphorylated (gray) and phosphorylated (red) Puf3-IDR$^{1–258}$. Puf3-IDR$^{1–258}$ was phosphorylated using a mixture of recombinant Sck1 and Pdk1 at a 1:5 molar ratio. Substantial chemical shift perturbations are observed upon phosphorylation, particularly backbone resonances of phosphorylated serines (black arrows). Inset: side-chain resonances of tryptophans. **b**, Scheme for producing differentially phosphorylated Puf3 for use in downstream experiments. **c**, Intact MS analysis of samples taken during the Puf3-IDR$^{1–376}$ phosphorylation time course. Deconvoluted spectra are shown with masses consistent with modal phosphorylation states shown above each time point (colored arrows). The black arrow (+18*) at 270 min shows maximum incorporation of 18 phosphates under these conditions, with a modal number of 13 phosphorylation sites (dark-red

arrow). **d**, Tryptophan indole peaks from $^1$H,$^{15}$N 2D-HSQC showing increasing chemical shift perturbation for the W229 side-chain resonance throughout the phosphorylation time course. These data suggest that structural changes occur in the DFW motif upon phosphorylation. **e**, $^{15}$N $R_2$ transverse relaxation rates of unphosphorylated (black bars) and phosphorylated (red bars) Puf3-IDR$^{109–258}$. This construct contains four of five phosphorylation sites identified by NMR (black arrows show positions of assigned phosphorylated residues). The increase in $R_2$ rates are consistent with a decrease in backbone dynamics upon phosphorylation. **f**, Deadenylation $t_{1/2}$ of Ccr4–Not with differentially phosphorylated full-length Puf3 proteins prepared according to **b**. IDR phosphorylation states were determined using intact MS, indicated as modal states (**c**). Experiments were performed in triplicate (open circles) and averaged (line).

phosphorylation, the $^{15}$N $R_2$ relaxation rates increase across the sequence, especially around the assigned phosphorylation sites. This increase is particularly pronounced around S156 and S207 and could indicate a decrease in local backbone mobility. This can be rationalized by the formation of an ion pair between the arginine side chain at position $i − 3$ and the phosphoserine in position $i$, resulting in a local increase in rigidity[26] (Fig. 4e and Extended Data Fig. 7b). Thus, our data suggest that phosphorylation modulates both the charges and the backbone dynamics of Puf3-IDR, leading to a reduced affinity for Ccr4–Not.

Next, we tested the effect of phosphorylation on the enzymatic activity of Ccr4–Not. We prepared differentially phosphorylated full-length Puf3 using three distinct phosphorylation timepoints (Fig. 4b) and assayed their abilities to stimulate Ccr4–Not activity using in vitro deadenylation assays. Increased phosphorylation of Puf3 does not affect RNA-binding capacity but reduces the ability of Puf3 to stimulate Ccr4–Not deadenylase activity (Fig. 4f and Extended Data Fig. 8). This reduction occurs in a graded manner where more phosphorylation results in slower deadenylation rates.

In summary, Puf3 can be sequentially phosphorylated, which reduces (but does not eliminate) its interaction with Ccr4–Not and its ability to stimulate deadenylation activity. This demonstrates that the in vitro enzymatic activity of Ccr4–Not toward specific RNAs can be tuned in a continuous manner. As Puf3 is also regulated by phosphorylation in vivo[24], it may tune deadenylation in a rheostat-like manner in cells.

## Multivalency and phosphoregulation are found in other Ccr4–Not-binding IDRs

Most known RNA adaptors bind Ccr4–Not through IDRs[1] (Supplementary Fig. 2). We hypothesized that multivalent binding and phosphoregulation are conserved features regulating deadenylation in higher eukaryotes. We, therefore, tested this with human RNA adaptors using a recombinant human CCR4–NOT complex[27].

Human PUM1 is a Pumilio RNA adaptor from the same family as *S. pombe* Puf3. IDRs in PUM1 and *Drosophila* PUM directly interact with the CNOT1, CNOT2 and CNOT3 subunits of CCR4–NOT through a region that is proximal to the Pumilio domain[12,13]. To test whether this IDR accounts for the stimulatory activity of human PUM1 in a fully reconstituted system, we purified N-terminal IDR deletions of PUM1 (Extended Data Fig. 9a). A truncation mutant containing a previously defined CCR4–NOT-interacting region (PUM1$^{\Delta587}$) retained the ability to stimulate CCR4–NOT but was ~4-fold less efficient than the full-length protein (Fig. 5a and Extended Data Fig. 9b,c). Other truncations showed that the length of the IDR in the PUM1 construct correlates with the ability to stimulate deadenylation. This graded response is consistent with a multivalent binding mechanism involving interaction regions throughout the IDR, similar to yeast Puf3.

The human RNA adaptor protein TTP (also known as ZFP36) selectively targets a set of proinflammatory mRNAs[28]. A conserved C-terminal SLiM[9] of TTP interacts with CNOT1 but additional regions in N-terminal and C-terminal IDRs have also been implicated in binding[21].

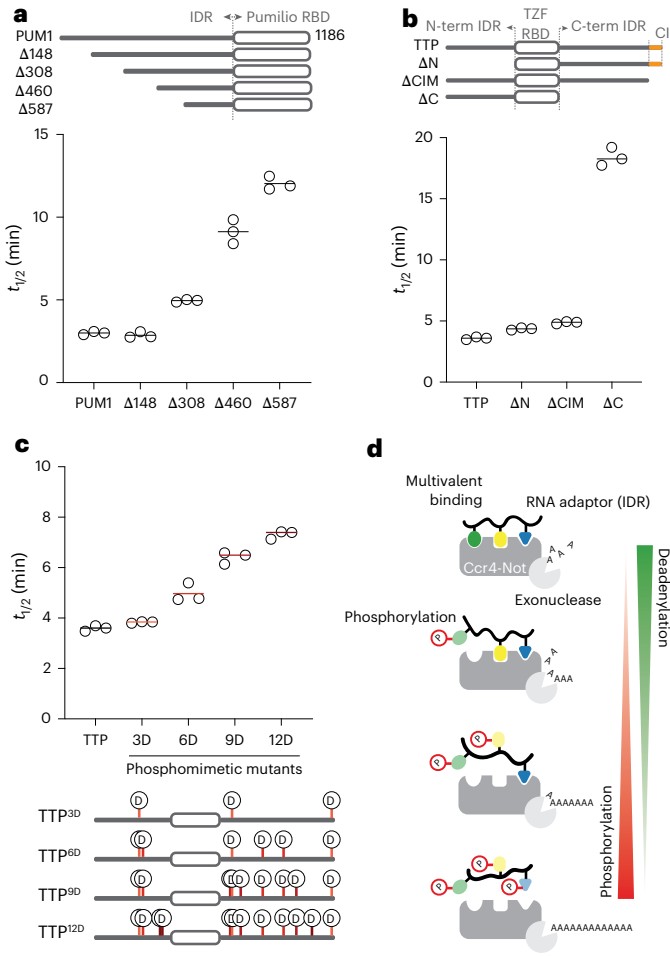

**Fig. 5 | Multivalency can tune human CCR4–NOT-mediated deadenylation with divergent RBPs. a**, Reconstitution of targeted deadenylation using recombinant full-length *H. sapiens* CCR4–NOT deadenylase complex and PUM1 constructs. Schematics of the PUM1 constructs are shown at the top. A 5′ fluorescently labeled RNA containing an upstream Pumilio recognition element followed by a 30-nt poly(A) tail was used as a substrate. Reactions were started by addition of the deadenylase complex and resulting products were resolved on denaturing urea PAGE gels. Experiments were performed in triplicate (open circles) and averaged (line). **b**, Deadenylation assays performed as in **a** but in the presence of TTP constructs and with a substrate RNA containing an upstream ARE. TTP contains both N-terminal and C-terminal IDRs around a central tandem zinc finger (TZF) RBD. A previously defined and structurally characterized CCR4–NOT interaction motif (CIM) is indicated in orange. **c**, Deadenylation assays performed with TTP constructs containing the indicated number of phosphomimetic substitutions. The locations of phosphomimetic substitutions are indicated in the schematic diagrams below. Assays were performed three times independently (open circles) and averaged (line). **d**, Model for multivalent interaction of RNA adaptors with CCR4–NOT and regulation of phosphorylation. RNA-binding proteins tether Ccr4–Not to specific RNAs using a multivalent mode of binding through IDRs. Interactions can be tuned through phosphorylation across extended IDRs.

TTP lacking the C-terminal SLiM shows a small but reproducible decrease in the ability to stimulate CCR4–NOT, whereas TTP lacking the entire C-terminal IDR shows a more substantial decrease (Fig. 5b and Extended Data Fig. 10a,b). Thus, additional elements in the IDR are important for maximal stimulation of deadenylation, likely by binding to CCR4–NOT through a multivalent mechanism.

TTP is known to be regulated by phosphorylation during the inflammatory response but the exact roles of many of the phosphorylation sites are unknown[29]. Given that these phosphorylation sites

are distributed throughout the IDR, we hypothesized that differentially phosphorylated TTP could exhibit similar behavior to Puf3 in tuning Ccr4–Not deadenylase activity. We purified full-length TTP variants with phosphomimetic substitutions (Extended Data Fig. 10c) and tested their ability to stimulate the deadenylation activity of CCR4–NOT. Interestingly, CCR4–NOT exhibits a graded response to TTP phosphomimetic mutations, similar to Puf3 phosphorylation (Fig. 5c and Extended Data Fig. 10d,e). However, phosphomimetic substitutions of TTP have a smaller impact on deadenylation activity compared to phosphorylation of Puf3. Thus, multiple RNA adaptors likely bind to multiple sites on Ccr4–Not using extended regions of their IDRs, which can be regulated in a tunable manner to control mRNA deadenylation.

## Discussion

In this study, we demonstrated that the yeast RNA adaptor protein Puf3 interacts with the Ccr4–Not deadenylase complex through a multivalent mechanism that can be regulated in a rheostat-like manner (Fig. 5d). Our data are also consistent with human RNA-binding proteins (PUM1 and TTP) using multiple interaction sites for Ccr4–Not binding and regulation. Specifically, multiple regions within the IDRs of RNA adaptors contribute to the ability to recruit Ccr4–Not to specific transcripts and to stimulate its deadenylation activity, possibly in a cooperative manner. The functional importance of such a mode of interaction lies in its regulatory potential; IDRs of RNA adaptors can be phosphorylated at specific sites, including within regions that bind to Ccr4–Not, thereby modulating their ability to regulate mRNA decay. In agreement with this, phosphorylation of Puf3 affects regulation of target transcripts in yeast[24] and phosphorylation of human TTP regulates mRNAs of the inflammatory response[28]. In the cell, additional factors could also bind to specific IDR segments in a regulated manner to further modulate effects on gene expression. For example, phosphopeptide-binding 14-3-3 proteins[30] could lead to a deviation in the graded response observed in our study. Thus, in our updated tethering model, multivalent interaction and PTMs can control the binding landscape between Ccr4–Not and specific transcripts. Given that regulation of poly(A) tail length is critical for mRNA half-life, phosphorylation can act as a rheostat to tune deadenylation rate and finely control RNA decay.

Previous studies showed that there are several distinct binding pockets on Ccr4–Not that interact with SLiMs. However, these studies generally investigated short peptides in isolation and distinct subcomplexes of Ccr4–Not, without examination of the direct impact on deadenylation activity[7,8,31,32]. Our strategy of reconstituting activity in vitro using full-length RNA adaptors and the complete Ccr4–Not complex shows that multiple regions of the IDR contribute to binding and individual SliMs are not sufficient for maximal stimulation of deadenylation. Another recent study also used a fully reconstituted system to show that TTP uses a multivalent binding mechanism to interact with multiple binding sites on human CCR4–NOT[33]. In that work, TTP phosphorylation did not substantially affect deadenylation directly but promoted interaction with poly(A)-binding protein PABPC1. Therefore, it is likely that TTP phosphorylation has multiple effects on deadenylation, dependent on which sites are modified.

In the multivalent interaction between RNA adaptors and Ccr4–Not, binding energy between a distinct SLiM and a Ccr4–Not binding pocket is likely small, consistent with a highly dynamic and shallow binding energy landscape[34]. This could have the advantage of faster kinetics at each binding site allowing greater accessibility to PTMs, a high capacity for regulation, and modularity where multiple RNA adaptors work with a single enzymatic complex.

Ccr4–Not binding pockets (for example, in Not1, Not3 and Not9) are highly conserved. In particular, many RNA adaptors interact with Not9, suggesting that Not9 is a general IDR interaction hub. We hypothesize that Not9 is evolutionarily constrained as it interacts with many different RNA adaptors. In contrast, IDRs normally have poor

sequence conservation but retain function across evolutionarily distant proteins[35,36]. The sequence plasticity of IDRs allows evolutionary tuning of processes—in this case, regulation of mRNA decay. Consistent with this, multivalency and IDRs are also important for interactions between the components of the decapping machinery[37]. The interplay involving different RNA adaptors, Ccr4–Not and the decapping machinery is currently not understood.

Our model for multivalent and rheostatic IDR interactions in mRNA decay has parallels with additional processes in gene expression. For example, multivalency is essential for high-affinity binding between transcription factor tandem activation domains (tADs) within Gcn4 and Mediator[16]. In this case, individual hydrophobic SLiMs in Gcn4 have low affinity and specificity for Mediator. Multivalent binding increases the affinity and specificity of complex formation but there is not a single distinct ordered state; the tADs are highly dynamic, with multiple binding conformations. It is possible that similar dynamic binding mechanisms occur between IDRs and Ccr4–Not. In another example, multisite phosphorylation of p53 promotes binding to CREB-binding protein/p300 (refs. 15,38) and decreases the interaction with Mdm2 (ref. 15) in a graded manner. Thus, sequential phosphorylation of IDRs also regulates effector binding in other processes within gene expression.

Overall, we demonstrate that RBPs mediate specificity by using a multivalent mechanism to interact with the Ccr4–Not complex that can be tuned by phosphorylation. Commonalities between deadenylation, decapping and transcription factors suggest that regulated multivalent interactions between IDRs in nucleic-acid-binding proteins and their effector complexes is likely a general strategy used to generate specificity within gene expression.

## Online content

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

## Methods

### Molecular cloning and synthetic DNA construct design

All DNA constructs described in this work were gene-synthesized to optimize codon usage for *Escherichia coli*. All PCR steps were performed using either Q5 or Phusion hot start high-fidelity polymerases and vectors were assembled isothermally using homology arms with NEBuilder HiFi DNA assembly master mix (New England Biolabs). Site-directed mutagenesis was performed by PCR using multiple overlapping fragments with primers incorporating substitutions before isothermal reassembly into vectors.

Gene-synthesized subunits for both the fission yeast and human Ccr4–Not complexes cloned into pACEBac vectors with polyhedrin promoter-based expression cassettes were described previously[27,39]. Cloning of *S. pombe* complexes was adapted to a modified BigBac system to streamline the introduction of point mutants[40,41]. Gene expression cassettes for Not1, Ccr4, Caf1 and Not9 in pACEBac1 vectors were PCR-amplified and assembled into a single modified pBig1A vector. Cassettes for Not2, Not3 and Not4 in pACEBac1 were similarly assembled into a single modified pBig1B vector. pBig1 vectors were then assembled into a single modified pBig2AB vector using Pme1 digestion and isothermal assembly. Assembled complex vectors were then transformed into DH10 EMBacY cells for transposition into recombinant baculoviruses[42].

*S. pombe* Puf3 expression vectors were previously described[17] and truncations were subcloned using specific PCR primers. *Homo sapiens* PUM1 constructs were subcloned from a gene-synthesized vector (GeneArt). Constructs for TTP including phosphomimetic mutants were gene-synthesized in expression vectors (Genscript). Kinase expression plasmids were subcloned using restriction digest from gene-synthesized plasmids (GeneArt).

### Recombinant protein expression and purification

**Ccr4–Not complexes.** Recombinant *S. pombe* and *H. sapiens* Ccr4–Not complex expression and purification were previously described[27,39] but further optimized for this study. Isolated Bacmids were transfected into adherent Sf9 cells (Oxford Expression Technologies, 600100) in six-well plates using FuGene Transfect HD (Promega). Viral supernatant was isolated 72 h after transfection and used immediately to infect expression cultures using a low-titer strategy that is necessitated by the instability of the large complex viruses. Then, 150 ml of Sf9 cells cultured in suspension in SF900-II SFM (Invitrogen) at $1.5 \times 10^6$–$2 \times 10^6$ cells per ml were infected with 1.5 ml of transfection supernatant. Cultures were then maintained at a cell density of $2 \times 10^6$–$3 \times 10^6$ by dilution with fresh medium until culture arrest ~72 h after infection. Cells were then cultured for a further 24–48 h before harvest by centrifugation at 1,000 rcf; pellets were gently washed with ice-cold PBS before flash-freezing and storage at −80 °C.

Frozen cell pellets were lysed by thawing in lysis buffer: 100 mM HEPES pH 8.0, 300 mM NaCl, 0.1% (v/v) Igepal CA-630, 2 mM DTT, protease inhibitor cocktail (Roche) and 0.4 mM PMSF. Lysate was cleared using ultracentrifugation at 200,000*g* for 30 min and filtered using 0.65 μM PVDF membranes (Millipore). Clarified lysate was bound in batch to 1 ml of Strep-Tactin Sepharose resin (IBA Lifesciences) per liter of input culture for 1.5 h. Beads were washed with 50 mM HEPES (pH 8.0), 150 mM NaCl, 2 mM magnesium acetate and 2 mM DTT before elution in buffer supplemented with 5 mM desthiobiotin (IBA Lifesciences). Eluate was directly loaded onto a HiTrap Q HP 5-ml column (Cytiva) and bound complexes were eluted using a gradient of 12 column volumes. Peak fractions were collected, pooled and injected onto size-exclusion chromatography (SEC) using Superose 6 prep-grade XK16/70 or 26/60 columns (Cytiva) equilibrated with 20 mM HEPES pH 8.0, 150 mM NaCl, 2 mM magnesium acetate and 0.5 mM TCEP. Peak fractions were pooled and loaded onto a Resource Q 1-ml column and eluted using a gradient of ten column volumes in 20 mM HEPES pH 8, ~300 mM NaCl, 2 mM magnesium acetate and 0.5 mM TCEP. This concentrated the complex to 2–5 mg ml⁻¹, which was then flash-frozen in liquid nitrogen for storage at −80 °C. For NMR experiments where complex at ~10 mg ml⁻¹ was required, the complex was supplemented with 10% (w/v) glycerol before further concentration using Amicon Ultra-4 (50-kDa molecular weight cutoff (MWCO)) centrifugal filter units.

Complex concentration was determined using ultraviolet (UV)–visible spectrophotometry. Stoichiometry and purity were determined at each purification step using NuPAGE 4–12% gradient SDS–PAGE gel analysis. Complex stability and dispersity were monitored using light scattering, mass photometry and differential scanning fluorimetry.

**Kinases.** Recombinant Sck1 and Ksg1 carrying N-terminal twin-Strep (SII) tags were overexpressed in Sf9 cells from baculoviruses transposed from pACEBac1 vectors. Bacmids were prepared and transfected into adherent Sf9 cells as for Ccr4–Not complexes and supernatant containing P1 virus isolated at 72 h after transfection. Virus was then amplified in suspension culture by infecting cells at a density of $1.5 \times 10^6$ cells per ml with 1:100 (v/v) P1 virus. Cell density was maintained by dilution with fresh medium until cultures arrested and viral P2 supernatant was collected at 72 h after infection. P2 virus was used to infect suspension expression cultures at $2.5 \times 10^6$ cells per ml. Cells expressing Ksg1 were harvested at 48 h after infection because of protein toxicity and those expressing Sck1 were harvested at 60 h after infection using centrifugation as detailed for Ccr4–Not complexes.

Frozen cell pellets were lysed by thawing in detergent as detailed for Ccr4–Not complexes. Lysate was clarified and kinases were purified using affinity purification as described for complexes above. Eluates were then directly injected onto a HiTrap Q HP 5-ml column (Cytiva) and bound protein was eluted using a gradient of 15 column volumes. Peak fractions were concentrated using Amicon Ultra-15 (30-kDa MWCO) centrifugal filter units.

**Pumilio constructs.** All *S. pombe* Puf3 and *H. sapiens* constructs for assays were overexpressed as maltose-binding protein (MBP) fusion proteins from a modified pMAL-c5x vector where the thrombin cleavage site was replaced with a 3C protease site. NMR constructs were overexpressed from modified pMAL-c5x vector carrying 3C-cleavable N-terminal MBP and C-terminal lipoyl tags. Puf3 protein constructs for anisotropy experiments were expressed from a modified pGEX vector carrying N-terminal GST and C-terminal lipoyl tags. Vectors were transformed into in-house prepared chemically competent BL21star (DE3) cells and positive colonies were directly used to inoculate expression cultures in 2×TY medium or minimal medium for isotopic labeling. MBP-tagged proteins for assays were expressed in 2×TY medium, grown to an optical density at 600 nm of 0.8–1 and induced with 0.5 mM IPTG for 4 h at 25 °C. GST-tagged fragments were induced overnight at 18 °C. Uniformly labeled proteins for NMR were expressed in M9 minimal medium (6 g l⁻¹ Na₂HPO₄, 3 g l⁻¹ KH₂PO₄ and 0.5 g l⁻¹ NaCl) supplemented with 1.7 g l⁻¹ yeast nitrogen base without NH₄Cl and amino acids (Sigma Y1251). In addition, 1 g l⁻¹ ¹⁵NH₄Cl and 4 g l⁻¹ unlabeled glucose were supplemented for ¹⁵N labeling. Unlabeled glucose was replaced with 3 g l⁻¹ [¹³C]glucose for ¹³C,¹⁵N double-labeled samples. Cells were harvested by centrifugation and stored at −80 °C before lysis.

Cell pellets were thawed in five volumes of lysis buffer containing 50 mM HEPES pH 8, 300 mM NaCl, 0.5 mM TCEP, 2 mM EDTA, 0.5 mM PMSF, protease inhibitor cocktail (Roche), 10 μM leupeptin and 2 μM pepstatin. Cells were lysed by sonication in 50-ml batches for 30 s in total with 30% amplitude using a 5-mm microtip on a VCX750 system (Sonics). Lysates were cleared by ultracentrifugation at 150,000*g* for 30 min before filtration through 0.65-μm filters. MBP-tagged protein was subsequently bound in batch for 1 h to 1 ml of amylose resin (New England Biolabs) per liter of input culture (typically 4 ml of total resin). The resin lysate suspension was then transferred to a 2.5 × 20 econo-column (Biorad) and resin was collected before the bed was

washed three times with 100 ml of wash buffer (50 mM HEPES pH 8, 300 mM NaCl, 0.5 mM TCEP, 2 mM EDTA and 5 μM leupeptin). Protein was then eluted using elution buffer (20 mM PIPES pH 6.8, 200 mM NaCl, 50 mM maltose, 0.01% (w/v) Brij-35 and 0.5 mM TCEP). For NMR constructs, tags were cleaved using 3C protease overnight at 5°.

Eluted proteins were immediately loaded onto a 5-ml HiTrap Heparin HP column equilibrated in 20 mM HEPES pH 6.8, 100 mM NaCl and 0.5 mM TCEP and bound protein was eluted using a linear gradient of 15 column volumes. Peak fractions were pooled and injected onto a Superdex 200 26/60 prep-grade size-exclusion column and proteins were eluted isocratically in 20 mM HEPES pH 7.5, 150 mM NaCl and 0.5 mM TCEP. For *S. pombe* Puf3 constructs, peak fractions were pooled and concentrated to ~5 mg ml$^{-1}$ using Amicon Ultra-15 (50-kDa MWCO) centrifugal filter units before flash-freezing for storage at −80 °C. For *H. sapiens* PUM1 constructs, pooled fractions from size exclusion were loaded onto a 1-ml RESOURCE S column equilibrated in 20 mM Bis–Tris pH 6, 50 mM NaCl, 0.5 mM TCEP and 10% (w/v) glycerol. Protein was then eluted using a linear gradient of 10 column volumes and peak fractions were flash-frozen for storage at −80 °C.

**TTP constructs.** All *H. sapiens* TTP constructs for assays were over-expressed as MBP fusion proteins from a modified pMAL-c5x vector where the thrombin cleavage site was replaced with a 3C protease site. Proteins were expressed in 2×TY medium and supplemented with 10 μM ZnSO$_4$ (final concentration) on induction with 0.5 mM IPTG for 4 h at 25 °C. TTP constructs were isolated and lysate was prepared as for Pumilio proteins. Proteins were bound in batch to amylose resin and washed three times with 100 ml of wash buffer (50 mM HEPES pH 8, 300 mM NaCl, 10 μM zinc acetate, 0.5 mM TCEP, 2 mM EDTA and 5 μM leupeptin). Bound proteins were then eluted with 20 mM PIPES pH 7, 200 mM NaCl, 0.5 mM TCEP, 1 μM zinc acetate and 0.1% Brij-35. Downstream purification steps of cation-exchange chromatography and SEC were then performed as for Pumilio proteins above.

**In vitro protein phosphorylation**
Purified Sck1, along with its activation loop kinase Pdk1, was produced as described above. Phosphorylated Puf3-IDR$^{1–258}$ and Puf3-IDR$^{109–258}$ for NMR analysis were produced using a dialysis method. Puf3-IDR constructs were purified until the first HiTrap Heparin HP column step. Peak fractions were pooled and mixed with a kinase master mix in a 5:1 molar ratio. Kinase master mix contained equimolar amounts of Sck1 and Pdk1 and was preincubated for 15 min with 5 mM adenosine triphosphate (ATP) and 20 mM MgCl$_2$ before mixing with Puf3. Then, 10-ml phosphorylation reaction mixtures were dialyzed 50-fold using Slide-A-Lyzer (10-kDa MWCO) G2 cassettes against 500 ml of phosphorylation buffer: 20 mM HEPES pH 7.4, 10 mM NaCl, 0.5 mM TCEP, 5 mM ATP (99% Roche) and 20 mM MgCl$_2$. Phosphorylation was typically allowed to proceed under dialysis for 18 h at 4 °C. To stop phosphorylation and purify the resulting phosphorylated protein, reaction mixtures were first applied to a HiTrap Heparin HP 5-ml column to remove kinase and unphosphorylated IDR. Flow through was collected and applied to a HiTrap Q HP 5-ml column and phosphorylated species were eluted using a gradient of 15 column volumes. Peak fractions were then injected onto a HiLoad Superdex 200 26/60 prep-grade column equilibrated in 20 mM HEPES pH 8, 100 mM NaCl and 0.5 mM TCEP. Phosphorylated IDRs were subsequently separated on a Capto HisRes Q 5/50 using a gradient of 50 column volumes before peak fractions were immediately dialyzed into NMR buffer.

Phosphorylation time-course experiments with full-length MBP–Puf3 were performed in 0.5-ml volumes in Slide-A-Lyzer MINI (10-kDa MWCO) dialysis devices in phosphorylation buffer: 20 mM HEPES pH 7.4, 50 mM NaCl, 4 mM ATP, 20 mM MgCl$_2$ and 0.5 mM TCEP. Puf3 concentration was adjusted to 32.5 μM and mixed with 6 μM kinase mixture (final concentration) to start reactions. Phosphorylation time courses were performed at room temperature under dialysis

with rapid stirring against a 50-fold excess volume of phosphorylation buffer. Next, 100-μl aliquots at desired time points were withdrawn and immediately desalted into 20 mM HEPES pH 8, 100 mM KCl, 0.5 mM EDTA and 0.5 mM TCEP using PD SpinTrap G-25 spin columns with a 25-μl stacker volume. Phosphorylated proteins were then immediately used in deadenylation activity assays, MS or RNA-binding assays.

**Analytical SEC and SEC–multiangle light scattering**
For analysis of the NMR sample containing 50 μM Puf3-IDR and 1 μM Ccr4–Not, 25 μl was injected through a capillary loop onto a Superose 6 increase 10/300 gl column equilibrated in 20 mM HEPES pH 7.5, 150 mM NaCl and 0.5 mM TCEP at 4 °C. Proteins were eluted isocratically at a flow rate of 0.4 ml min$^{-1}$.

Light-scattering experiments were run using an Agilent 1260 Infinity II liquid chromatography (LC) system with an online Dawn Helios II system (Wyatt) with a QELS+ module (Wyatt) and an Optilab rEX differential refractive index detector (Wyatt). One hundred microliters of Ccr4–Not sample at 0.1 mg ml$^{-1}$ was autoinjected onto a Superose 6 increase 10/300gl column (GE Healthcare) run at 0.5 ml min$^{-1}$. Molecular weights were calculated in ASTRA software (Wyatt), with a BSA control sample at 2 mg ml$^{-1}$ used for normalization and calibration of delay volumes.

**Deadenylation assays**
Assays using *S. pombe* Ccr4–Not were performed as previously described with some modifications[17,39]. Substrate RNA was synthesized with a 5′ 6-FAM fluorophore and contained a Pumilio response element (PRE) embedded within a 20-nt non-poly(A) region, upstream of a 30-nt poly(A) tail: AACUGUUCCUGUAAAUACGCCAG(A)$_{30}$. Recombinant Ccr4–Not complexes were diluted to either 0.5 μM (10× stock) for targeted assays (in the presence of an RNA adaptor) or 1 μM for assays alone (no RNA adaptor in Extended Data Fig. 6c) in 20 mM HEPES pH 8, 300 mM NaCl, 2 mM magnesium acetate and 0.5 mM TCEP. Puf3 constructs were diluted to 2.5 μM (10× stock) in 20 mM HEPES pH 8, 150 mM KCl and 0.5 mM TCEP. Assays were performed in a total volume of 100 μl in a thermally controlled block at 22 °C. RNA (final concentration 200 nM) was preincubated with 250 nM Puf3 construct for 15 min in deadenylation buffer (20 mM PIPES pH 6.8, 10 mM KCl, 2 mM magnesium acetate and 0.1 mM TCEP). For reactions where the MBP tag was removed, 10× protein stocks were incubated with 0.02 μg μl$^{-1}$ 3C protease (final concentration) on ice for 60 min before preincubation with substrate RNA. Reactions were then started by the addition of 10× Ccr4–Not stock to a final concentration of 50 nM (100 nM for control reactions in Extended Data Fig. 6c). Then, 7.5 μl of sample at indicated time points was withdrawn and mixed with an equivalent volume of loading dye (95% formamide, 4 mM EDTA and 0.01 w/v bromophenol blue). Deadenylated species were resolved on denaturing TBE (Tris–borate–EDTA) 20% polyacrylamide (19:1) gels containing 7 M urea and run at 400 V in 1× TBE running buffer for 30 min. Gels were visualized using an Amersham Typhoon 5 system (Cytiva) using a 488-nm excitation laser and 525-nm 20-nm bandpass emission filter.

Assays using *H. sapiens* CCR4–NOT were performed as previously described but with conditions optimized for the human complex. First, 10× CCR4–NOT complex and 10× RNA-binding protein stocks were prepared at 0.5 μM and 2.5 μM, respectively, in the same buffers as for the *S. pombe* complex above. Assays with TTP were performed with an RNA containing an upstream ARE element instead of the PRE element, also containing a 30-nt poly(A) tail and a 5′ 6-FAM fluorophore for visualization: AAUCAUCCUUAUUUAUUACCAUU(A)$_{30}$. RNA at a final concentration of 200 nM was preincubated with indicated 250 nM RNA-binding protein construct in deadenylation buffer optimized for the human complex (20 mM HEPES pH 7.8, 80 mM NaCl, 2 mM magnesium acetate and 0.1 mM TCEP final concentration). Reactions were started with the addition of 10× CCR4–NOT complex; samples

were withdrawn, the reaction was stopped and species were resolved on denaturing PAGE gels as with *S. pombe* complex assays described above.

Assays were cropped and contrast adjusted linearly in Photoshop (Adobe) before being imported into ImageJ[62]. Complete lane profiles were integrated and intensity values obtained for both total substrate and deadenylated substrate per lane and time point. Relative deadenylated fractions were calculated and plotted in Prism (GraphPad) as a function of time. Points were joined with straight lines to interpolate the time where half the poly(A) tail was removed ($t_{1/2}$). Assays and quantification were performed three times for every construct.

## Pulldown assays

Binding assays were preequilibrated in a total volume of 50 µl, containing 1 µM recombinant Ccr4–Not complex and 6 µM MBP-tagged Puf3 fragment (equating to 25 µg of immobilized protein for the full-length construct) in binding buffer: 20 mM HEPES pH 7.5 mM, 150 mM NaCl and 0.5 mM TCEP. Amylose resin was washed in ultrapure water and binding buffer, resuspended as a 50% slurry and 30 µl was added to the binding reactions (corresponding to a final bead volume of 15 µl). Reactions were incubated for 15 min at room temperature and washed three times with 500 µl of cold binding buffer; bound proteins were eluted with 40 µl of 2× LDS loading dye (Invitrogen). Complexes were resolved using 4–12% NuPAGE gradient SDS–PAGE gels in MOPS-based running buffer and Coomassie-stained for 2 h to overnight with Instant-Blue (Abcam). Gels were destained with successive washes of ultrapure water and 5% (v/v) ethanol and 7% (v/v) acetic acid before imaging. To quantify fractional binding, the intensities of the Not1 subunit band were integrated in ImageJ and plotted relative to full-length Puf3.

## Fluorescence anisotropy

Binding of *S. pombe* Puf3 constructs to Ccr4–Not were assayed using the complex as a fluorescent probe. Ccr4–Not was site-specifically labeled with FITC on the N terminus of the Not9 subunit using a YbbR tag[43]. CoA–fluorescein conjugates were synthesized from CoA and fluorescein maleimide before reverse-phase (RP) high-performance LC (HPLC) purification, lyophilization and LC–MS validation. An in-frame YbbR tag was introduced at the N terminus of Not9 through PCR and assembled into a vector containing the full Ccr4–Not complex for baculovirus mediated overexpression. Complex was purified as above and labeled by incubation of Ccr4–Not with a fivefold molar excess of CoA–fluorescein and 1:100 Sfp enzyme (prepared in house) in 20 mM HEPES pH 8, 150 mM NaCl, 0.5 mM TCEP and 5 mM MgCl$_2$. Excess dye and Sfp were then removed using SEC and the complex was further purified by anion-exchange chromatography as detailed above.

Binding was measured by titration of increasing concentrations of Puf3 against 25 nM labeled Ccr4–Not (final concentration) in binding buffer: 20 mM HEPES pH 8, 100 mM NaCl and 0.5 mM TCEP, in a total volume of 20 µl using 384-well black round-bottom low-volume nonbinding surface assay microplates (Corning). Fluorescence anisotropy was measured using a PHERAstar Plus (BMG Labtech). Binding curves were fitted using a quadratic function, taking ligand depletion into account, to calculate apparent dissociation constants.

For RNA binding, fluorescently labeled RNA oligonucleotides were used, containing a PRE element embedded within a 20-nt RNA. Twofold protein dilution series were prepared at 10× concentration in dilution buffer (20 mM HEPES pH 7.5, 100 mm NaCl and 0.5 mm TCEP). Protein was incubated at room temperature for 30 min with 0.1 nm 5′ 6-FAM-labeled RNA (synthesized by Integrated DNA Technologies) in buffer containing 20 mM HEPES pH 7.5, 150 mM NaCl and 0.1 mM TCEP, in a total volume of 100 µl using a 384-well low-flange black flat-bottom nonbinding surface microplate (Corning). Fluorescence anisotropy was measured using a PHERAstar Plus (BMG Labtech). Dissociation constants were estimated using a quadratic function in Prism 6.0 (GraphPad software). Error bars indicate the s.d. of two technical replicates.

## CLMS

Sulfo-NHS-Diazirine (Sulfo-SDA)-based UV-induced crosslinking was performed both on *S. pombe* Ccr4–Not complex alone and with a 1.5-fold molar excess of Puf3. Sulfo-SDA stocks were freshly dissolved in ultrapure water at 100 mM before being added to a final concentration of 0.5, 0.1 and 0.05 mM in three separate crosslinking reactions with a total of 100 µg of complex adjusted to 0.2 µg µl$^{-1}$. Reactions were incubated on ice for 30 min before the diazirine moiety was activated by irradiation under a 365-nm UV lamp for 20 min. Bands corresponding to crosslinked complex in the 0.5 mM samples were excised from SDS–PAGE gels. Lower-concentration (0.1 and 0.05 mM) samples were then mixed and precipitated using ice-cold acetone; proteins were pelleted and dried.

The precipitated protein samples were resolubilized in digestion buffer (8 M urea in 100 mM ammonium bicarbonate) to an estimated protein concentration of 1 mg ml$^{-1}$. Disulfide bonds in the samples were reduced by adding DTT to a final concentration of 5 mM, followed by incubation at room temperature for 30 min. The free sulfhydryl groups were then alkylated by adding iodoacetamide to a final concentration of 15 mM and incubating at room temperature for 20 min in the dark. After alkylation, the excess iodoacetamide was quenched by adding DTT to a final concentration of 10 mM.

Next, the protein samples were digested with LysC at a 50:1 (w/w) protein-to-protease ratio at room temperature for 4 h. The samples were then diluted with 100 mM ammonium bicarbonate to reduce the urea concentration to 1.5 M. Trypsin was added at a 50:1 (w/w) protein-to-protease ratio to further digest the proteins for 15 h at room temperature. The resulting peptides were desalted using C18 StageTips[44].

For each sample, the resulting peptides were fractionated using SEC to enrich for crosslinked peptides[45]. Peptides were separated using a Superdex 30 Increase 3.2/300 column (GE Healthcare) at a flow rate of 10 µl min$^{-1}$. The mobile phase consisted of 30% (v/v) acetonitrile and 0.1% trifluoroacetic acid. The first seven peptide-containing fractions (50 µl each) were collected and the first two fractions were combined. The solvent was removed using a vacuum concentrator and the fractions were then analyzed by LC–MS/MS.

LC–MS/MS analysis was performed using an Orbitrap Fusion Lumos Tribrid MS instrument (Thermo Fisher Scientific) connected to an Ultimate 3000 RSLCnano system (Dionex, Thermo Fisher Scientific). Each SEC fraction was resuspended in 1.6% (v/v) acetonitrile and 0.1% (v/v) formic acid and analyzed by LC–MS/MS. Each SEC fraction was analyzed in duplicate by LC–MS/MS. Peptides were injected onto a 50-cm EASY-Spray C18 LC column (Thermo Scientific), operated at 50 °C. Mobile phase A consisted of water with 0.1% (v/v) formic acid and mobile phase B consisted of 80% (v/v) acetonitrile with 0.1% (v/v) formic acid. Peptides were loaded and separated at a flow rate of 0.3 µl min$^{-1}$. Separation was achieved using linear gradients, increasing from 2% to 55% B over 92.5 min. The gradient slope was optimized for each SEC fraction. Following separation, the B content was ramped to 95% over 2.5 min. Eluted peptides were ionized by an EASY-Spray source (Thermo Scientific) and introduced directly into the MS instrument.

The MS data were acquired in data-dependent mode. In each 2.5-s acquisition cycle, the full-scan mass spectrum was recorded in the Orbitrap at a resolution of 120,000. Ions with a charge state of 3+ to 7+ were isolated and fragmented using higher-energy collisional dissociation. For each isolated precursor, a collision energy of 26%, 28% or 30% was applied, depending on the $m/z$ and charge state of the precursor[46]. The fragmentation spectra were then recorded in the Orbitrap at a resolution of 60,000. Dynamic exclusion was enabled with a single repeat count and a 30-s exclusion duration.

For MS raw data processing, MS2 peak lists were generated using the MSConvert module in ProteoWizard (version 3.0.11729). Precursor and fragment $m/z$ values were recalibrated. Crosslinked peptides were identified using the xiSEARCH software (version 2.0)

(https://www.rappsilberlab.org/software/xisearch)[47]. The data from the apo Ccr4–Not complex and Ccr4–Not + Puf3 samples were analyzed separately. Peak lists were searched against the protein sequences of the subunits of the *S. pombe* Ccr4–Not complex and, for the Ccr4–Not + Puf3 sample, the sequence of Puf3 was included. The reversed protein sequences were used as decoys for error estimation.

The following parameters were applied during the search: MS accuracy at 3 ppm, MS2 accuracy at 5 ppm, enzyme specificity set to trypsin with full tryptic cleavage, up to three missed cleavages allowed and allowance for up to two missing monoisotopic peaks. Carbamido-methylation on cysteine was set as a fixed modification, while oxidation on methionine was set as a variable modification. SDA was specified as the crosslinker, with reaction specificity for lysine, serine, threonine, tyrosine and protein N termini for the NHS ester end and any amino acid for the diazirine end. SDA loop links and hydrolyzed SDA on the diazirine end were also considered as variable modifications.

Crosslinked peptide candidates from each sample were filtered using xiFDR (version 2.2.betaB)[48,49] with a requirement of a minimum of three matched fragment ions per crosslinked peptide, including at least two ions containing a crosslinked residue. A false discovery rate of 1% at the residue-pair level was applied, with the 'boost between' option enabled. The MS proteomics data were deposited to the ProteomeXchange Consortium through the PRIDE[50] partner repository with the dataset identifier PXD055147.

### Intact MS

Proteins were first desalted and buffer-exchanged into ammonium acetate (50 mM) and formic acid (0.1% v/v) using Microcon MWCO filters (Millipore). Denatured proteins were washed five times before being diluted to a concentration of approximately 2 µg µl$^{-1}$. Proteins were analyzed by direct infusion intact MS. All analysis was carried out on a Synapt G2-Si (Waters) using a pulled borosilicate glass emitter coupled to a nanoelectrospray ion source in positive ion mode. The data were acquired using 5-s scan rates and the spectra were processed using UniDec[51] deconvolution.

### NMR spectroscopy

All NMR samples were dialyzed against 2 L of the following buffers before data collection. For backbone assignment experiments, the samples were prepared in 20 mM PIPES pH 6.8, 50 mM NaCl, 2 mM TCEP and 0.02% sodium azide. For Ccr4–Not binding experiments, the samples were dialyzed into 20 mM PIPES pH 6.8, 150 mM NaCl, 2 mM TCEP and 0.02% sodium azide. For phosphorylation time courses, the samples were prepared in 20 mM PIPES pH 6.8, 50 mM NaCl, 2 mM TCEP, 0.02% sodium azide, 1 mM ATP and 2 mM MgCl$_2$. Time courses were started with the addition of 1:5 kinase. All NMR samples were supplemented with 5% (v/v) D$_2$O (final concentration).

NMR experiments were acquired at 278 K with Bruker TopSpin 3.6.5 acquisition software, using an in-house Bruker 800-MHz Avance III spectrometer, equipped with a triple-resonance TCI cryogenic probe. Access to a Bruker 950-MHz Avance III spectrometer equipped with a TCI cryogenic probe (Medical Research Council (MRC) Biomedical NMR Center, Francis Crick Institute) provided experiments with increased resolution and sensitivity. Lastly, for the acquisition of $^{13}$C-detected experiments, a Bruker Avance III 700-MHz spectrometer with TXO heteronuclear detection cryogenic probe was used. Unless otherwise specified, 75 µM samples in standard 5-mm NMR tubes were used, apart from Ccr4–Not binding experiments where 50 µM labeled sample was used. All data were referenced directly ($^1$H) or indirectly ($^{15}$N,$^{13}$C) according to the water signal.

Backbone resonances of all Puf3-IDR constructs (Puf3-IDR$^{1–258}$, Puf3-IDR$^{109–258}$, phosphorylated Puf3-IDR$^{1–258}$ and phosphorylated Puf3-IDR$^{109–258}$) were obtained with band-selective excitation short transient–transverse relaxation-optimized spectroscopy triple-resonance experiments correlating Cα, Cβ and CO resonances:

BT-HNCO, BT-HN(CA)CO, BT-HNCACB, BT-HN(CO)CACB (Bruker pulse sequence library)[52] and additional HSQC-based (H)N(COCA)NNH experiments. All BT three-dimensional 3D experiments were acquired at 800-MHz $^1$H resonance frequency with 2,000 × 40 ($^{15}$N) × 64 ($^{13}$C) complex points, a $^{15}$N sweep width of 21 ppm and offset of 119 ppm, respectively. The (H)N(COCA)NNH experiments were acquired with 2,000 × 80 ($^{15}$N) × 80 ($^{15}$N) complex points and identical $^{15}$N parameters.

To complete the assignment of backbone imino resonances of prolines and for residues that underwent rapid solvent exchange, we obtained additional $^{15}$N,$^{13}$C resonances using $^1$H start versions of $^{13}$C-detected CON, CACON and CANCO experiments (Bruker pulse sequence library).

All 3D datasets were collected with nonuniform sampling at 10–25% and processed with NMRPipe[53] and qMDD[54] for compressed sensing reconstruction. Data analysis was achieved in Bruker Topspin 4.1.1, NMRFAM-Sparky 1.47 and POKY[55,56]. Backbone resonances were assigned semiautomatically in MARS[57] and curated manually using in-house scripts.

Secondary chemical shifts were calculated on the basis of differences in experimental Cα and Cβ chemical shifts from calculated random coil chemical shifts[58] (https://www1.bio.ku.dk/english/research/bms/sbinlab/randomchemicalshifts2) accounting for experimental temperature, salt concentration and pH.

Binding studies with Ccr4–Not used fHSQC experiments (Bruker pulse sequence library) acquired at 950 MHz. To measure the extent of binding with and without phosphorylation for the Puf3-IDR$^{109–258}$ construct, peak intensities were scaled to the most flexible and least perturbed residues (200–201, GD) and expressed as a ratio of the peak heights of the bound and free forms, respectively.

$^{15}$N transverse ($R_2$) relaxation rates were measured at 950-MHz $^1$H resonance frequency using pseudo-3D experiments (hsqct2etf3gp-sitc3d, Bruker pulse sequence library) with an initial recovery delay of 5 s including temperature compensation to avoid sample heating effects (NS = 48 and 2,000 × 64 complex points in the $^1$H,$^{15}$N planes). Transverse relaxation times of 8.48 (repeated twice), 16.96, 33.92, 50.88, 67.84, 101.76, 135.68, 169.6, 203.52, 685 237.44 and 271.36 ms were acquired in an interleaved fashion and relaxation rates were obtained from the intensities in each plane by fitting to an exponential decay either in NMRFx[59] within the RING NMR[60] plugin or after processing in TopSpin 4.1.1 (Bruker) in combination with a peak height analysis in NMRFAM-Sparky (version 1.4)[56].

### Structural predictions

Residues 1–129 and 160–291 of Puf3 were tiled in 50-aa sections with a 3-aa step size. Structural predictions of tiles were generated with AlphaFold2 Multimer in ColabFold 1.5.2 in complex with the following *S. pombe* Ccr4–Not subunits: Not9 in complex with the Not9-binding domain of Not1 (residues 1098–1318), Not1 N-terminal TTP-binding domain (residues 640–821) and NOT module (Not1 residues 1568–2100, Not2 residues 74–306 and Not3 471–640). We chose to use AlphaFold2 rather than AlphaFold3 because of the latter's tendency to hallucinate order in disordered regions. pLDDT values for Puf3 residues were extracted for every model and plotted by Puf3 residue using the Bokeh package. Structural predictions were rendered in PyMol.

### Reporting summary

Further information on research design is available in the Nature Portfolio Reporting Summary linked to this article.

### Data availability

All assay gels used for quantification were deposited to figshare (https://doi.org/10.6084/m9.figshare.29961968)[61]. NMR data were deposited to the Biological Magnetic Resonance Data Bank with accession number 52282. MS data were deposited to ProteomeXchange with

primary accession code PXD055147. All unique materials are available on request from the corresponding author upon completion of a standard materials transfer Agreement. All other data supporting the findings of this study are available from the corresponding author on reasonable request. Source data are provided with this paper.

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

## Acknowledgements

We thank P. Alcón (MRC Laboratory of Molecular Biology (LMB)) for advice on protein labeling, J. G. Shi (MRC-LMB) for support with baculovirus, M. Su (MRC-LMB) for help with RP-HPLC, S. McLaughlin (MRC-LMB) for advice with biophysics, H. Fagarasan (MRC-LMB) for proofreading, C. Russo (MRC-LMB) and M. Peet (MRC-LMB) for helpful discussions, and all members of the L.A.P. group for useful discussions and advice. This work was supported by the MRC as part of UK Research and Innovation (MRC file reference number MC_U105192715; L.A.P.), the European Union's Horizon 2020 research and innovation program (European Research Council Consolidator grant 725685; L.A.P.), Gates Cambridge (L.K.D.), the Deutsche Forschungsgemeinschaft (DFG, German Research Foundation) under Germany's Excellence Strategy EXC 2008 (390540038, UniSysCat9; J.R.), the Wellcome Trust through a Discovery Award (227434; J.R.) and core funding for the Wellcome Center for Cell Biology (203149; J.R.). NMR was supported by the Francis Crick Institute through provision of access to the MRC Biomedical NMR Center. The Francis Crick Institute receives its core funding from Cancer Research UK (CC1078), the UK Medical Research Council (CC1078) and the Wellcome Trust (CC1078).

## Author contributions

J.A.W.S. and L.A.P. conceptualized the study and designed the experiments. J.A.W.S. purified the proteins and performed the in vitro assays, with help from G.C.L. and S.A. C.W.H.Y., J.A.W.S. and S.M.V.F. performed NMR. L.K.D. and J.A.W.S performed AlphaFold screens. Z.A.C., T.M., L.S., F.J.O. and J.R. performed MS experiments. J.A.W.S. and L.A.P. wrote the paper and prepared the figures with input from all authors. L.A.P. supervised the project.

## Competing interests

The authors declare no competing interests.

## Additional information

**Extended data** is available for this paper at https://doi.org/10.1038/s41594-025-01688-1.

**Correspondence and requests for materials** should be addressed to Lori A. Passmore.

reports are available. Primary Handling Editor: Melina Casadio, in collaboration with the *Nature Structural & Molecular Biology* team.

**Rights retention statement** This work was supported by the Medical Research Council, as part of United Kingdom Research and Innovation (also known as UK Research and Innovation) [MRC file reference number MC_U105192715]. For the purpose of open access, the MRC Laboratory of Molecular Biology has applied a CC BY public copyright licence to any Author Accepted Manuscript version arising.

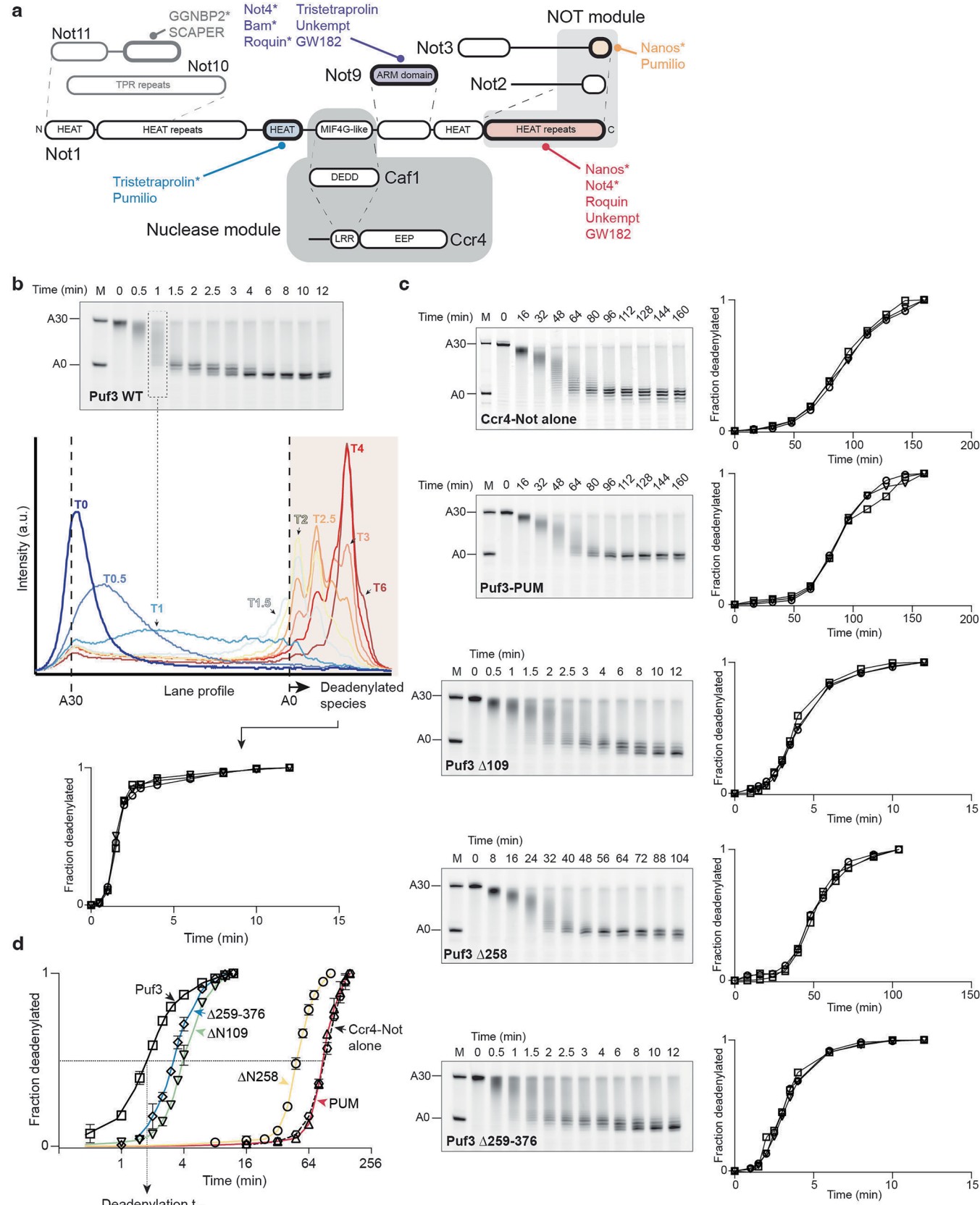

**Extended Data Fig. 1 | See next page for caption.**

**Extended Data Fig. 1 | Architecture of Ccr4-Not and deadenylation assay setup.** (**a**) Schematic diagram showing known interactions between domains of Ccr4-Not (domain diagrams) and RNA-binding protein (colored text) short linear motifs (SLiMs). In the domain diagrams, structured domains are depicted as rounded rectangles and linkers or disordered regions are shown as connecting lines. All core subunits of the complex assemble around the Not1 scaffold protein (interacting regions indicated with dashed lines) except Ccr4, which is bridged via Caf1. Colored domains contain known interaction surfaces with SLiMs. Interactions between SLiMs and domains of Ccr4-Not that have been studied through X-ray crystal structures are labelled with asterisks. (**b**) Representative deadenylation assay and densitometry analysis. Wild-type Puf3 (WT) was incubated to form a 1:1 complex with a 5′ fluorescently labelled substrate RNA, containing a Pumilio response element (PRE) (UGUAAAUA) in a 20-nt region upstream of a 30-nt poly(A) tail. Puf3-bound RNAs were then incubated with recombinant Ccr4-Not complex and fractions were taken at indicated time points. Products were resolved on denaturing PAGE before imaging (top). Intensity profiles across each lane were integrated (middle) and the fraction of deadenylated RNA was quantified for each time point. Data were then plotted as a function of time (bottom), enabling quantitation of deadenylation rate. Assays were performed in triplicate with quantification shown for all replicates. Straight lines are drawn between data points for easier visualization. M, molecular weight marker. (**c**) Deadenylation assays with Ccr4-Not alone (top) and indicated Puf3 truncations, showing a representative gel and quantitation in triplicate. Domain diagrams for the Puf3 constructs are shown in Fig. 1a. (**d**) Summary graph of assays shown in (c) but plotted on a $\log_2$ time scale. Open shapes for each construct show the average of 3 replicate assays with error bars indicating standard deviation. The $t_{1/2}$ values used for plotting deadenylation activities were interpolated from the plot.

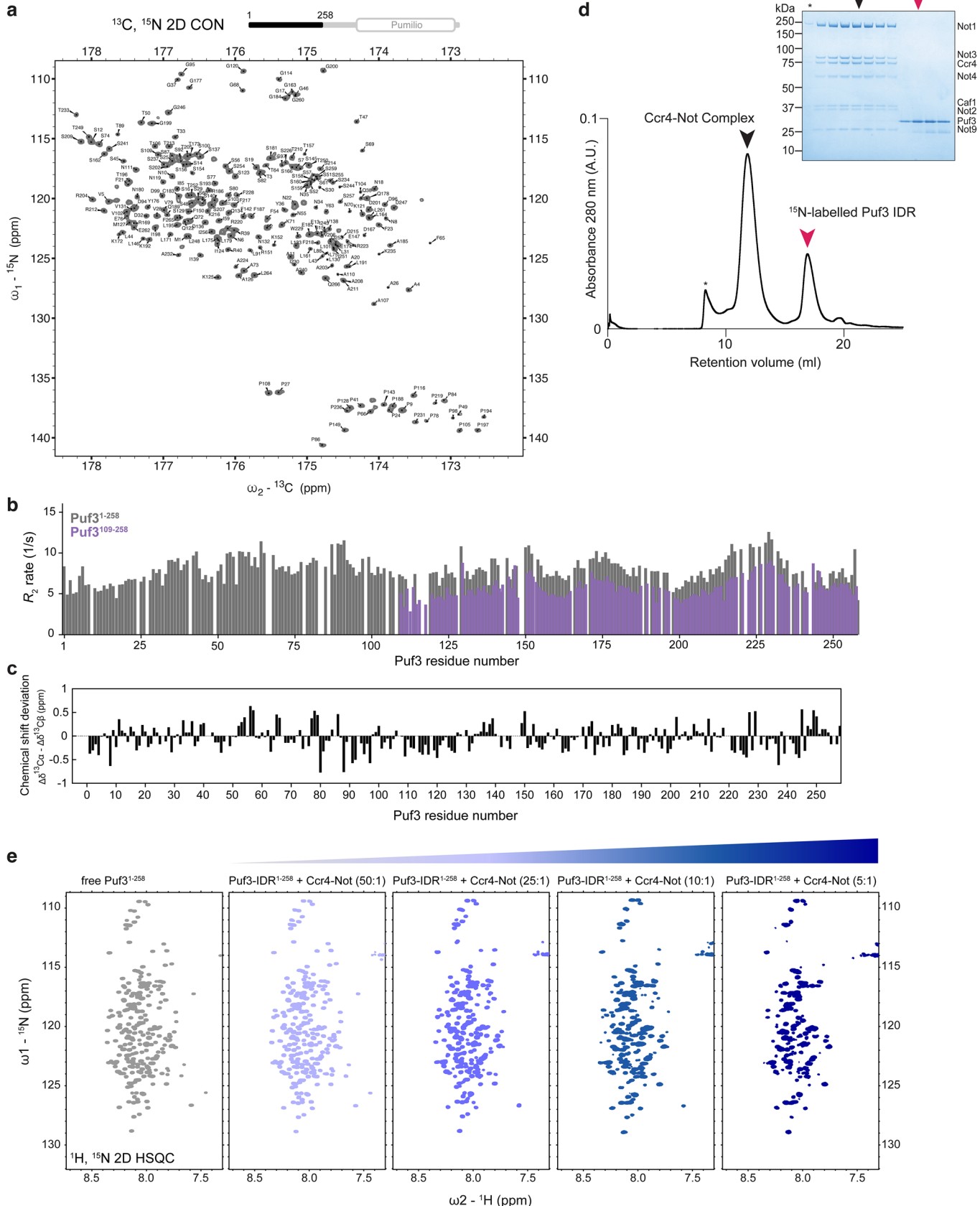

**Extended Data Fig. 2 | See next page for caption.**

**Extended Data Fig. 2 | Mapping the interaction of Ccr4-Not on Puf3.** (**a**) $^{13}$C,$^{15}$N 2D CON spectrum of 75 µM Puf3-IDR$^{1-258}$. (**b**) $R_2$ transverse relaxation rates of Puf3-IDR$^{1-258}$ (black) and Puf3-IDR$^{109-258}$ (purple). The lower global $R_2$ rates for the shorter construct are consistent with a shorter overall correlation time but there are no clear effects on the local structure from the N-terminal portion of the longer construct. (**c**) Secondary chemical shift analysis ($\Delta\delta^{13}C\alpha$ - $\Delta\delta^{13}C\beta$) for Puf3-IDR$^{1-258}$. Positive values would suggest residual helicity within that region.

These data show that Puf3-IDR$^{1-258}$ lacks substantial secondary structure. (**d**) Size exclusion chromatography analysis of 50 µM Puf3-IDR$^{1-258}$ in the presence of 1 µM Ccr4-Not. SDS-PAGE analysis of indicated peak fractions shows no stable complex formation under these experimental conditions (not repeated independently). (**e**) $^1$H,$^{15}$N 2D-HSQC spectra of 50 µM Puf3-IDR$^{1-258}$ in the presence of the indicated molar ratios of unlabeled Ccr4-Not. Substantial exchange broadening is observed with increasing concentrations of Ccr4-Not.

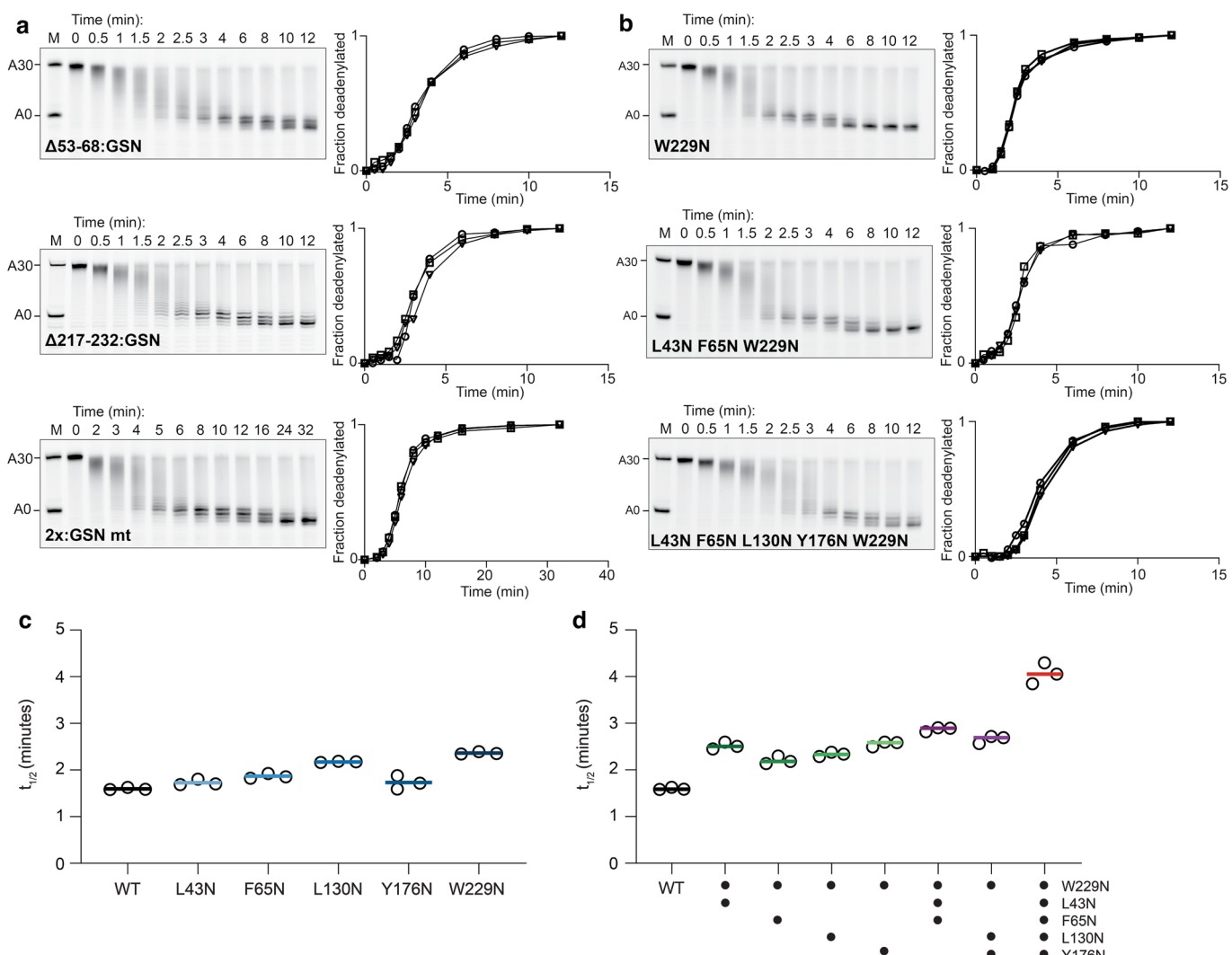

**Extended Data Fig. 3 | Deadenylation and binding assays performed in the presence of Puf3 mutants.** (**a**, **b**) Representative assay gels and quantitation of 3 independent assays for indicated mutant Puf3 proteins, with residues 53-68, 217-232 or both substituted for a GSN linker (**a**) or point mutants (**b**). (**c**, **d**) Deadenylation half-lives ($t_{1/2}$) of Puf3 stimulated Ccr4-Not activities of (**c**) single point mutants and (**d**) combined point mutants. Lines show the calculated mean from three independent replicates (open circles).

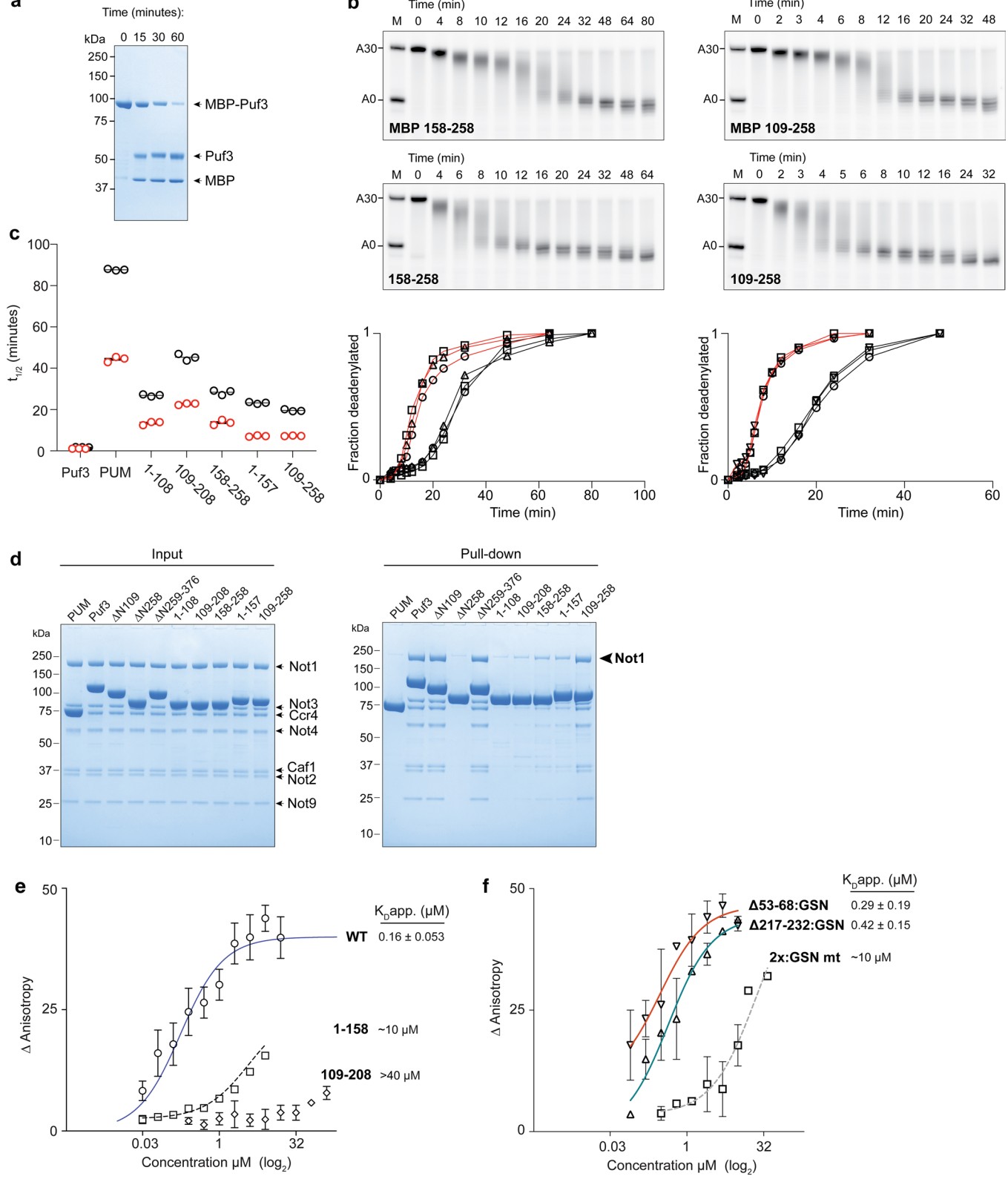

**Extended Data Fig. 4 | See next page for caption.**

**Extended Data Fig. 4 | Binding and activity assays of Puf3 IDR truncations.**
(**a**) SDS-PAGE analysis of cleavage of N-terminal MBP tag with 3C protease on ice from purified Puf3 protein over time. The Puf3-MBP mixture after 60 min cleavage was used for assays in (**b**). (**b**) Representative assay gels and quantitation for indicated truncated Puf3 proteins (related to Fig. 2b.) Plots show activities with MBP tagged protein (black lines) and with proteins after cleavage of MBP (red lines) to test for the effects of the tag. (**c**) Summary plot of deadenylation half-lives ($t_{1/2}$) of indicated truncation constructs with MBP tagged protein (black lines) and with proteins after cleavage of MBP (red lines). Assays were performed three times independently (open circles) and averaged (line). All constructs show systematic ~2-fold faster deadenylation rates without an MBP tag, possibly due to steric effects. The relative deadenylation rates and thus the conclusions of the experiment are not affected. (**d**) Recombinant purified MBP tagged Puf3 truncation constructs were immobilized on Amylose resin and tested for their ability to bind full-length recombinant Ccr4-Not complex. Left: Representative SDS-PAGE gel of experimental inputs with Ccr4-Not subunits labelled. Right: Representative SDS-PAGE gel of pull-downs. Experiments were performed in triplicate and quantified in Fig. 3c according to the relative band intensity of the Not1 subunit (arrow in bold). (**e**, **f**) Fluorescence anisotropy based analysis of Puf3 binding to labelled Ccr4-Not. Increasing concentrations of the indicated Puf3 constructs were titrated into 25 nM FITC-labelled Ccr4-Not. Binding isotherms were fitted with a quadratic model taking account of bound ligand. Symbols represent the mean of 2 replicates with error bars showing standard deviation. Calculated apparent $K_D$ values (± standard deviation) are shown. Puf3 binding to Ccr4-Not is generally correlated to its ability to promote deadenylation by Ccr4-Not. However, there is not always a direct correlation for several reasons: 1) The concentrations of proteins used in the assays (250 nM Puf3 construct, 50 nM Ccr4-Not) may result in limitations when comparing lower affinity peptides. 2) Truncations may result in a different orientation of bound RNA with respect to the active sites of Ccr4-Not, causing differences in activity not related to IDR-Ccr4-Not affinity but due to steric effects. 3) Given that Ccr4-Not deadenylates RNA even in the absence of Puf3, one would not expect a linear correlation between binding affinity and activity (ie. a peptide with 100-fold less binding affinity would not have 100-fold less activity).

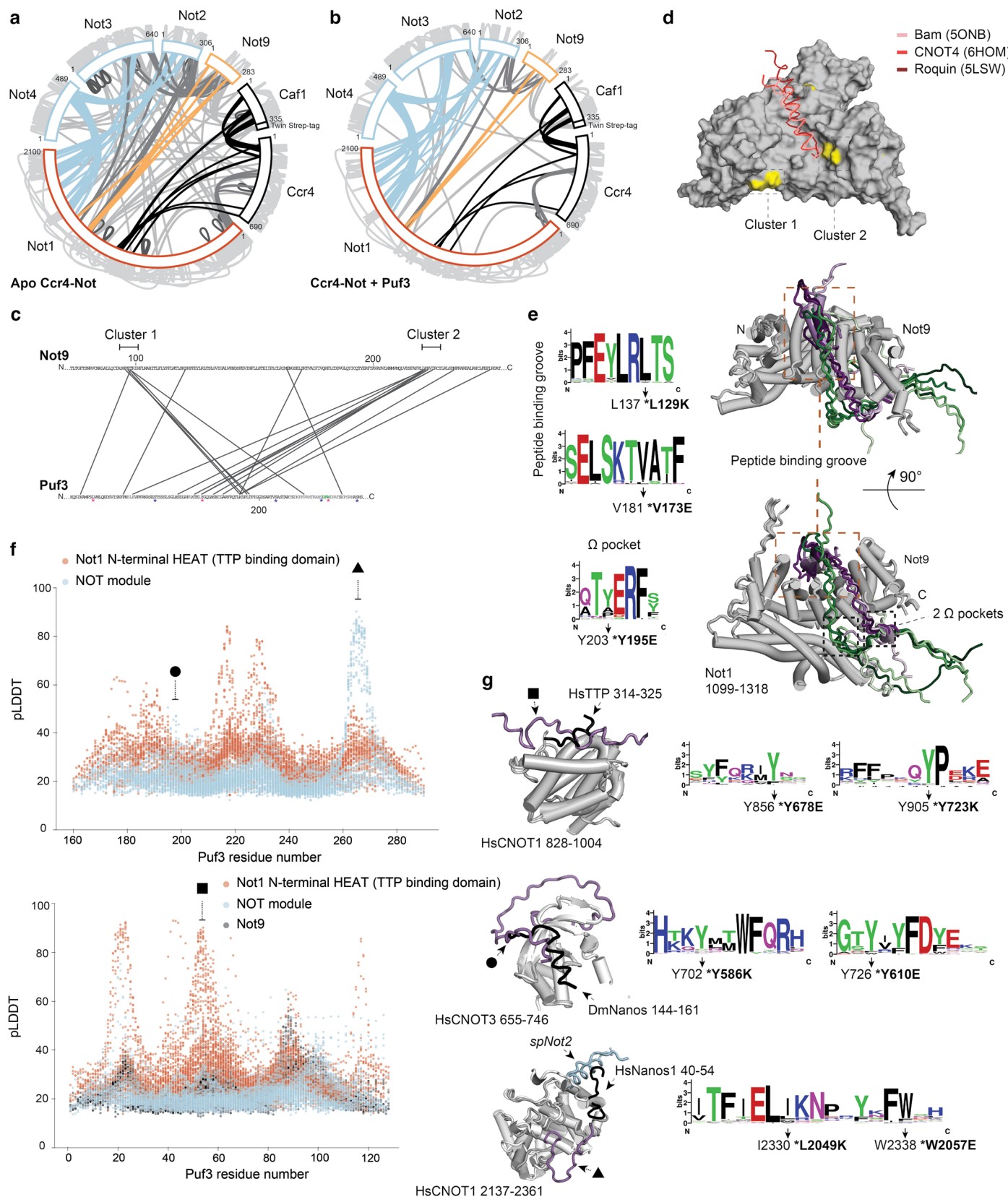

**Extended Data Fig. 5 | See next page for caption.**

**Extended Data Fig. 5 | Crosslinking mass spectrometry analysis,** *in silico* **prediction and evolutionary conservation of Puf3 - Ccr4-Not interactions.**
(**a**, **b**) Circular representation of crosslinking mass spectrometry analysis (CLMS) of *Schizosaccharomyces pombe* Ccr4-Not, alone (**a**) and in the presence of 1.5-fold molar excess of Puf3 (**b**). Recombinant full-length complexes were subjected to UV-activated Sulfo-SDA CLMS. Proteins and crosslinks are colored according to the subunit architecture of Ccr4-Not, with interprotein crosslinks that cannot be explained by the known complex architecture in grey within the circle. Intra-residue links are shown in dark grey. Intra-protein crosslinks are displayed around the outside in light grey. To allow comparison, Puf3 crosslinks have been omitted in panel (**b**). (See Fig. 3a for CLMS diagram with Puf3). These data are consistent with no large scale conformation changes in Ccr4-Not on binding Puf3. (**c**) Detailed view of crosslinks between Not9 and Puf3 mapped onto segments of the protein sequences showing two distinct clusters on the Not9 subunit. The DFW motif is highlighted in green. Due to poor coverage of trypsin cleavage sites, we could not detect peptides from all regions of the Puf3 IDR. Asterisks mark phosphorylated residues (blue) and residues mutated in this study (magenta). (**d**) Mapping of CLMS Not9 cluster 1 and 2 from panel (**c**) (yellow) onto the surface of the predicted *Schizosaccharomyces pombe* Not9 structure in proximity to the peptide binding groove. Peptides from Bam, CNOT4 and Roquin co-crystal structures with *Homo sapiens* CNOT9 are overlaid and PDB IDs are indicated. (**e**) Right: Cartoon representation of structural predictions of Not9 module in complex with Puf3 as also shown in Fig. 3c as a surface representation. Ten Puf3 tiles containing the highest pLDDT scores centred around the DFW motif are overlayed. Predicted structures engage at least one of the two previously-described tryptophan-binding pockets. Some predictions engage both pockets, despite the Puf3 tiles containing a single tryptophan residue. We therefore more

broadly define these pockets as as aromatic (Ω) pockets, since they are also predicted to bind residues other than tryptophan (for example phenylalanine). Predicted structures are displayed in two orientations, showing that multiple Puf3 tiles engage both the peptide binding groove and Ω-pocket(s). Left: Evolutionary conservation of CNOT9 binding sites depicted as sequence logos, generated using WebLogo from consensus sequence alignments representing a broad range of eukaryotic clades. Residues mutated in fission yeast Not9 (also see Extended Data Fig. 6) in this study are marked in bold next to the corresponding residue from the crystal structures of *H. sapiens* CNOT9 shown in d. (**f**) Plot of predicted local distance difference test (pLDDT) values per Puf3 residue from AlphaFold2-based screens with the indicated Ccr4-Not subcomplexes and tiles from Puf3 residues 160-291 (top) or residues 1-129 (bottom). Also see Fig. 3b. Filled shapes denote which tiles are shown as representative structures in panel (**g**). (**g**) Analysis of conserved Not1 and Not3 hydrophobic SLiM-binding pockets, as observed in co-crystal structures and AlphaFold2-based screens. Structures are shown on the left and sequence conservation plots (as in panel (**e**)) are shown on the right. Top: Overlay of a predicted structure of a Puf3 fragment from the AlphaFold2-based screen (dark purple) bound to part of the N-terminal HEAT repeat domain of Not1, with a co-crystal structure of human CNOT1 bound to human (Hs) TTP (black; PDB 4J8S). Middle: CNOT3 from the co-crystal structure between HsCNOT1 (not shown), HsCNOT3 (grey cartoon) and *Drosophila* Nanos (DmNos; black) (PDB 5FU7) with overlay of representative AlphaFold2-based interaction screen hit of Puf3 (light purple) bound to Not3. Bottom: Analysis of a hydrophobic pocket at the C-terminus of HsCNOT1 bound to HsNos (black; from crystal structure PDB ID: 4CQO), with overlay of representative tiled peptide from Puf3 interaction screen (light blue). In all cases, the residues lining the binding pocket of Ccr4-Not subunits are highly conserved.

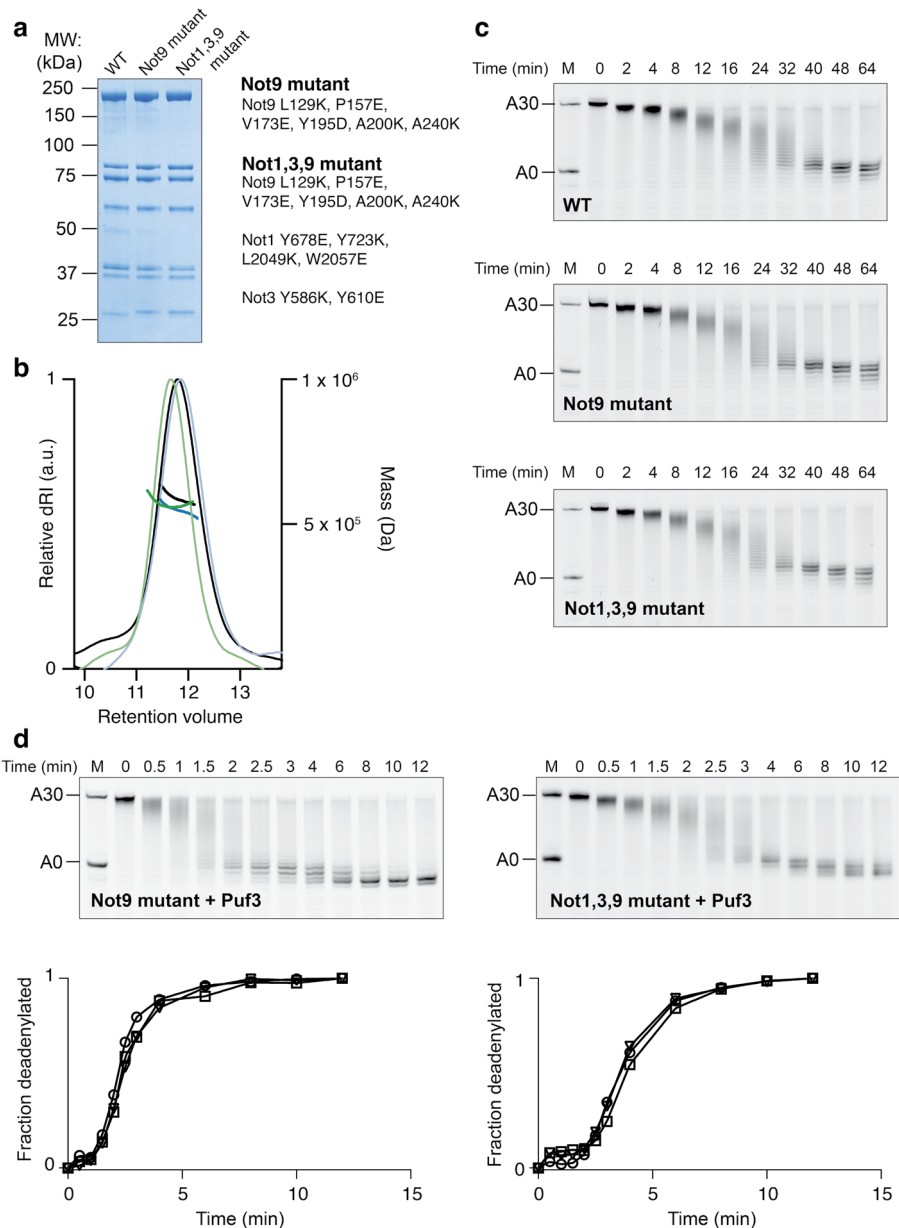

**Extended Data Fig. 6 | Deadenylation activities of mutant Ccr4-Not complexes. (a)** Coomassie-stained SDS-PAGE analysis of purified recombinant Ccr4-Not complexes carrying indicated mutations in conserved SLiM binding sites, as detailed in Extended Data Fig. 5. Purification of mutant complexes were each performed once. M, molecular weight marker. **(b)** Size exclusion chromatography coupled to multiangle light scattering analysis of mutant complexes, showing that all complex variants are intact and monodisperse. 100 µl of each complex was injected on a Superose 6 increase 10/300 gl column at 200 nM and analyzed using an inline coupled light scattering instrument.

Peak traces (WT: Black, Not9 mutant: blue, Not1,3,9 mutant: green) show the normalized differential refractive index with the calculated mass plotted as lines (right axis). **(c)** Deadenylation assays of 100 nM Ccr4-Not mutant complexes alone (without Puf3) on a substrate RNA containing a PRE and a 30-nt poly(A) tail, showing that there are no differences in intrinsic activity. **(d)** Representative deadenylation assays and corresponding densitometry analysis of assays performed in triplicate of indicated Ccr4-Not mutant complexes in the presence of WT Puf3 (related to Fig. 3d).

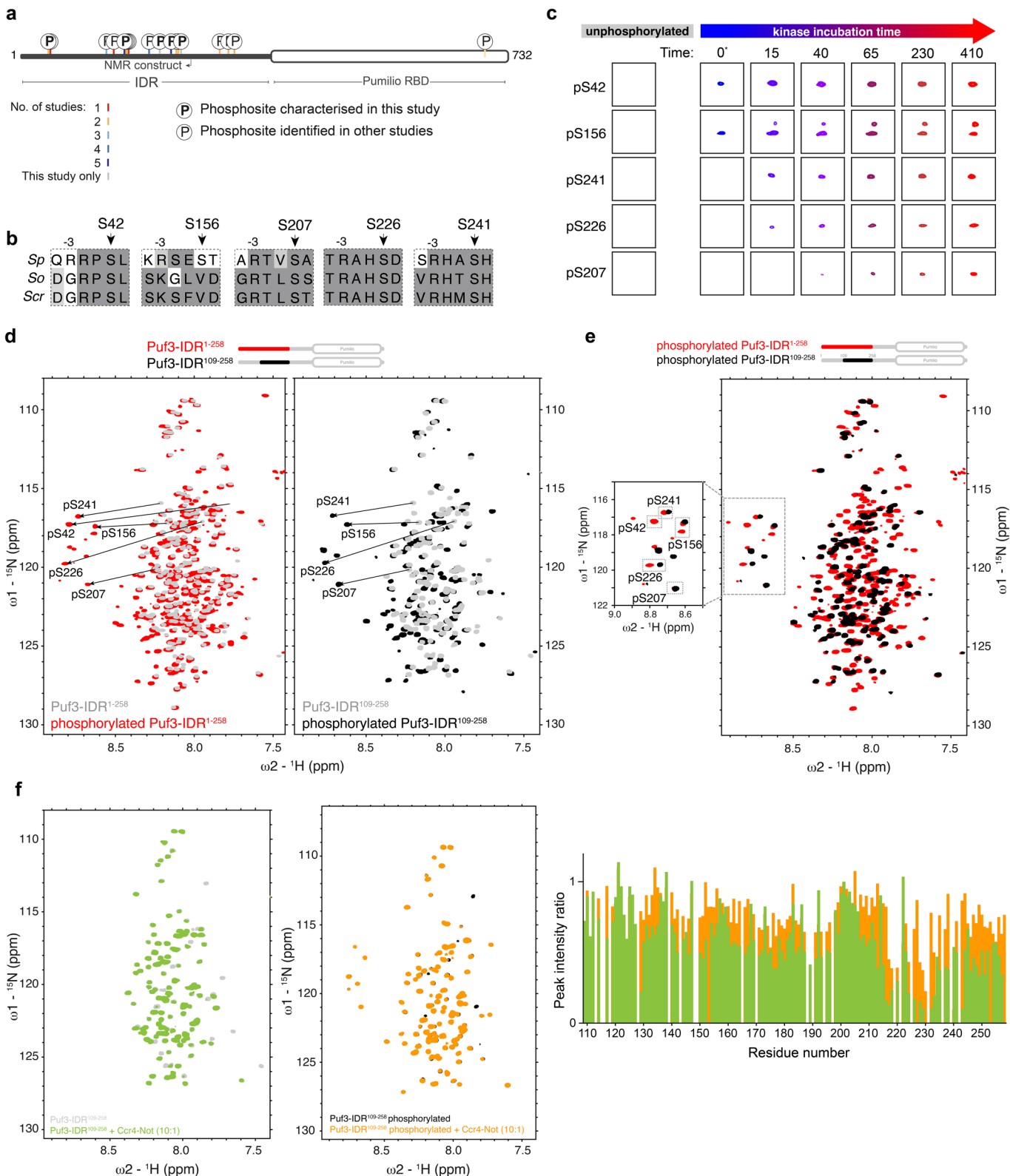

**Extended Data Fig. 7 | See next page for caption.**

**Extended Data Fig. 7 | Progressive phosphorylation of the Puf3 IDR attenuates association with Ccr4-Not. (a)** Schematic of annotated Puf3 phosphorylation sites identified in this study and previous phosphoproteomics studies (as curated in PomBase[23]). Color code shows the number of studies (including this one) identifying a particular site. Sites in bold are characterized using NMR spectroscopy in this study. (**b**) Alignments of *Schizosaccharomycetes* Puf3 sequences around the assigned phosphorylation sites. *Sp, S. pombe; So, S. octosporus; Scr, S. cryophilus.* Arg is often found in the -3 position. (**c**) Selected regions of 2D $^1$H $^{15}$N HSQC spectra of phosphorylation time courses. Addition of kinases leads to appearance of peaks in a distinct temporal pattern. The kinase was added at 0 min (asterisk), but includes the acquisition time, so the spectrum shows changes at S42 and S156 due to rapid phosphorylation at these sites under these conditions. Note that splitting of the S156 peak is likely due to substoichiometric phosphorylation at an unassigned neighboring site. (**d**) Overlays of 2D $^1$H $^{15}$N HSQC spectra of non-phosphorylated and phosphorylated Puf3-IDR$^{1-258}$ and Puf3-IDR$^{109-258}$. (**e**) Overlay of 2D $^1$H $^{15}$N HSQC spectra of phosphorylated Puf3-IDR$^{1-258}$ and Puf3-IDR$^{109-258}$ samples used for assignment. Inset shows assigned peaks of phospho-serine residues. (**f**) Binding of Ccr4-Not to non-phosphorylated and phosphorylated Puf3-IDR$^{109-258}$. Left: Spectral overlays of $^1$H,$^{15}$N 2D-HSQC of 50 μM Puf3-IDR$^{109-258}$ shown for unphosphorylated (grey and green) and phosphorylated (black and yellow) proteins with and without 5 μM unlabeled Ccr4-Not. Right: Peak intensity ratios of non-phosphorylated (green) and phosphorylated (yellow) Puf3-IDR$^{109-258}$ in the presence of Ccr4-Not from $^1$H,$^{15}$N 2D-HSQC plotted from spectra on left.

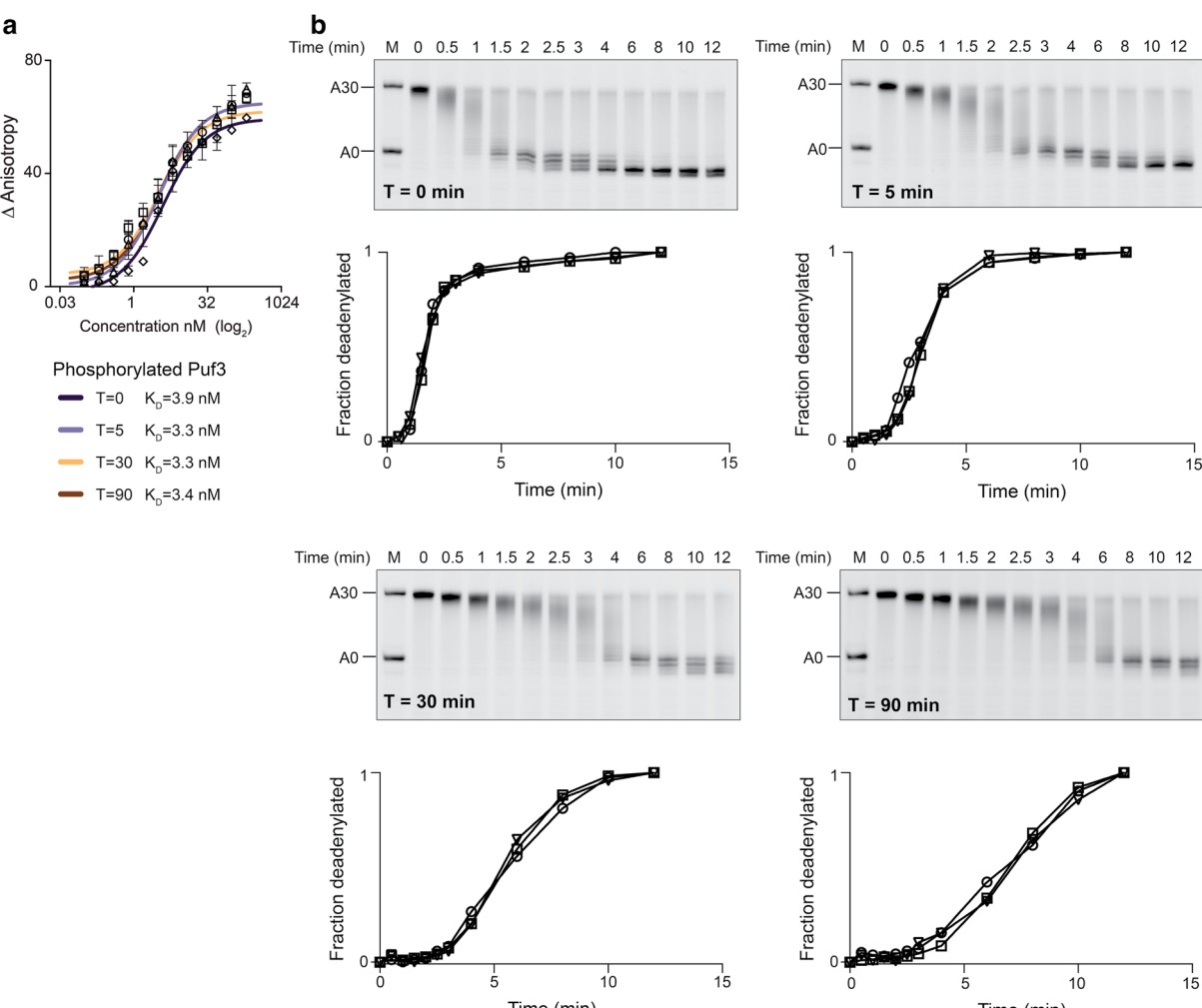

**Extended Data Fig. 8 | RNA binding and deadenylation activity assays of phosphorylated Puf3.** (**a**) Fluorescence anisotropy of 5′6-FAM labelled PRE-containing RNA, titrated against increasing concentrations of MBP-Puf3 phosphorylated for the indicated time. Data were fitted with a quadratic binding equation to calculate the dissociation constants shown below. Data points are the mean of at least 3 technical replicates with error bars showing standard deviation. (**b**) Representative deadenylation assays and densitometry analysis of data plotted in Fig. 4f. The ability of full-length MBP-tagged Puf3 phosphorylated for the indicated times to stimulate the deadenylation activity of Ccr4-Not was analyzed according to Extended Data Fig. 1b. Three replicate assays are shown in each of the plots.

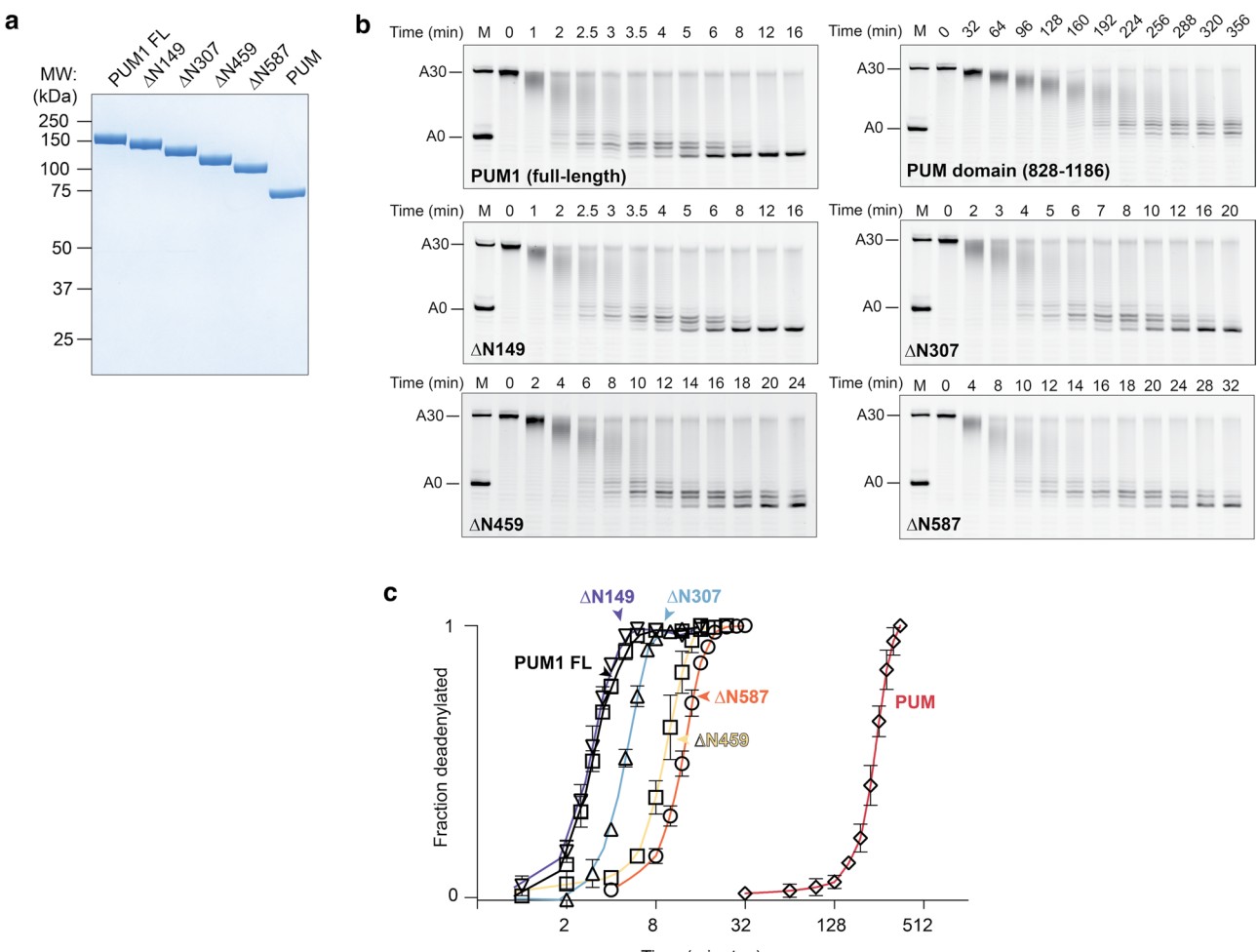

**Extended Data Fig. 9 | Reconstitution of human Pumilio (PUM1) mediated targeted deadenylation.** (**a**) SDS-PAGE analysis of purified PUM1 truncations. Truncations were designed based on previous studies defining conserved regions of the IDR. PUM, Pumilio domain alone. Purifications of mutant proteins were each performed once. (**b**) *In vitro* deadenylation assays probing the ability of recombinant MBP-tagged PUM1 protein constructs to accelerate the exonucleolytic shortening of poly(A) tails from a synthetic RNA substrate by full-length recombinant human CCR4-NOT complex. MBP-tagged PUM1 constructs were preincubated to form a 1:1 complex with a 5′ 6-FAM labelled substrate RNA containing a Pumilio recognition element (PRE) (UGUAAAUA)

in a 20-nt upstream region and a 30-nt poly(A) tail. Reactions were started by the addition of 50 nM CCR4-NOT complex, samples withdrawn at indicated time points, and reaction species separated on denaturing PAGE gels before imaging. One representative gel is shown for each construct. (**c**) Quantification of deadenylation assays (from panel **b**). Open shapes for each construct show average of 3 replicate assays. Error bars show standard deviation. The Δ587 construct containing ~250 predicted unstructured residues proximal to the Pumilio RNA-binding domain retains the ability to stimulate removal of the poly(A) tail by ~10-fold.

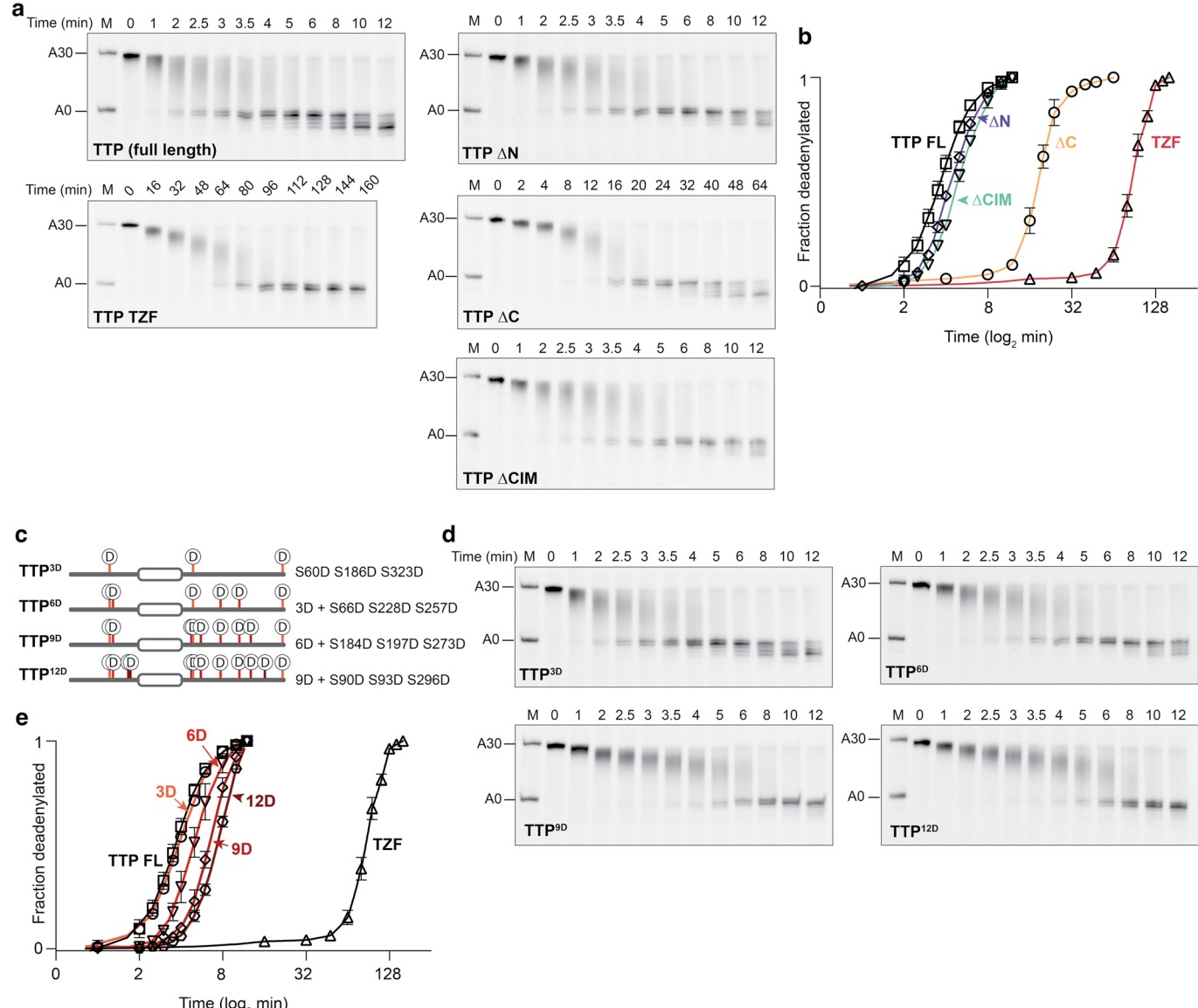

**Extended Data Fig. 10 | Reconstitution of targeted deadenylation with human Tristetraprolin (TTP) phosphomimetic variants.** (**a**) *In vitro* deadenylation assays testing the ability of recombinant human MBP-tagged TTP protein IDR deletion constructs to accelerate the deadenylation activity of full-length recombinant *Homo sapiens* CCR4-NOT complex. Indicated MBP-tagged TTP constructs were preincubated to form a 1:1 complex with a 5′ 6-FAM labelled substrate RNA containing an AU-rich element (ARE) (UUAUUUAUU) in a 20-nt upstream region and a 30-nt poly(A) tail. Reactions were started by the addition of 50 nM CCR4-NOT complex and samples withdrawn at indicated time points before reaction species were separated on denaturing PAGE gels and imaged. A representative gel is shown for each construct. (**b**) Quantification of deadenylation assays for constructs shown in panel (**a**). Open shapes for each construct show the average of 3 replicate assays with error bars showing standard deviation. The structurally characterized interaction between the CIM motif and NOT1 subunit is not sufficient to fully account for the stimulatory activity of TTP, consistent with previous data[10]. (**c**) Phosphomimetic TTP constructs used in this study. All phosphorylation sites have previously been characterized using either phosphoproteomics MS or phosphopeptide mapping[29]. Point mutant proteins at indicated positions (mutated to Asp) were cloned, overexpressed and purified to homogeneity for assays performed in panel **d**. (**d**) Representative *in vitro* deadenylation assay gels performed as in (**a**) but with indicated TTP phosphomimetic constructs. (**e**) Quantification of assays in (**d**) as described in (**b**). Whilst a graded response is observed the magnitude of the inhibition is small, which could be a result of the use of phosphomimetics.

# Reporting Summary

## Statistics

For all statistical analyses, confirm that the following items are present in the figure legend, table legend, main text, or Methods section.

| n/a | Confirmed | |
|---|---|---|
| ☐ | ☒ | The exact sample size (*n*) for each experimental group/condition, given as a discrete number and unit of measurement |
| ☐ | ☒ | A statement on whether measurements were taken from distinct samples or whether the same sample was measured repeatedly |
| ☒ | ☐ | The statistical test(s) used AND whether they are one- or two-sided *Only common tests should be described solely by name; describe more complex techniques in the Methods section.* |
| ☒ | ☐ | A description of all covariates tested |
| ☒ | ☐ | A description of any assumptions or corrections, such as tests of normality and adjustment for multiple comparisons |
| ☐ | ☒ | A full description of the statistical parameters including central tendency (e.g. means) or other basic estimates (e.g. regression coefficient) AND variation (e.g. standard deviation) or associated estimates of uncertainty (e.g. confidence intervals) |
| ☒ | ☐ | For null hypothesis testing, the test statistic (e.g. *F*, *t*, *r*) with confidence intervals, effect sizes, degrees of freedom and *P* value noted *Give P values as exact values whenever suitable.* |
| ☒ | ☐ | For Bayesian analysis, information on the choice of priors and Markov chain Monte Carlo settings |
| ☒ | ☐ | For hierarchical and complex designs, identification of the appropriate level for tests and full reporting of outcomes |
| ☒ | ☐ | Estimates of effect sizes (e.g. Cohen's *d*, Pearson's *r*), indicating how they were calculated |

*Our web collection on statistics for biologists contains articles on many of the points above.*

## Software and code

Policy information about availability of computer code

| Data collection | TopSpin (Bruker Version 4.1.1). |
|---|---|
| Data analysis | Adobe Photoshop (Release 25.10). Image J (Version 1.53K). Prism (GraphPad Version 10.3). MARS (BMG Version 5.02). ProteoWizard (Version 3). xiSEARCH (Version 2.0). xiFDR (Version 2.2). UniDec (Version 6). TopSpin (Bruker Version 4.1.1). NMRFAM-Sparky (Version 1.47). Mars (Version 1.0). POKY (build 08/29/2024j). NMRFx (Version 11.4) |

For manuscripts utilizing custom algorithms or software that are central to the research but not yet described in published literature, software must be made available to editors and reviewers. We strongly encourage code deposition in a community repository (e.g. GitHub). See the Nature Portfolio guidelines for submitting code & software for further information.

## Data

Policy information about availability of data

All manuscripts must include a data availability statement. This statement should provide the following information, where applicable:

- Accession codes, unique identifiers, or web links for publicly available datasets
- A description of any restrictions on data availability
- For clinical datasets or third party data, please ensure that the statement adheres to our policy

All assay gels used for quantification have been deposited on figshare [doi:10.6084/m9.figshare.29961968, ref. 63]. NMR data have been deposited in the BMRB database with the accession number 52282. Mass spectrometry data have been deposited in ProteomeXchange with the primary accession code PXD055147

[doi:10.6019/PXD055147, ref. 64]. All unique materials are available on request from the corresponding author with completion of a standard Materials Transfer Agreement. Source data have been provided in Source Data. All other data supporting the findings of this study are available from the corresponding author on reasonable request.

## Research involving human participants, their data, or biological material

Policy information about studies with [human participants or human data](). See also policy information about [sex, gender (identity/presentation), and sexual orientation]() and [race, ethnicity and racism]().

| | |
|---|---|
| Reporting on sex and gender | N/A |
| Reporting on race, ethnicity, or other socially relevant groupings | N/A |
| Population characteristics | N/A |
| Recruitment | N/A |
| Ethics oversight | N/A |

Note that full information on the approval of the study protocol must also be provided in the manuscript.

## Field-specific reporting

Please select the one below that is the best fit for your research. If you are not sure, read the appropriate sections before making your selection.

☒ Life sciences ☐ Behavioural & social sciences ☐ Ecological, evolutionary & environmental sciences

For a reference copy of the document with all sections, see [nature.com/documents/nr-reporting-summary-flat.pdf](nature.com/documents/nr-reporting-summary-flat.pdf)

## Life sciences study design

All studies must disclose on these points even when the disclosure is negative.

| | |
|---|---|
| Sample size | This is not applicable to our study. |
| Data exclusions | Data were not excluded. |
| Replication | All assays were performed three times, as indicated in the text. |
| Randomization | Randomization is not relevant to the assays and biochemical experiments performed in this study. For biochemical assays, the same protein stock was used for a given experiment. Replicates of the experiment show the results are reproducible and are not subject to the researchers' bias. |
| Blinding | Blinding is not relevant to the experiments presented in this study. Biochemical data were collected and processed identically under the same experimental conditions in an unbiased manner, and sample information did not lead to bias on any sample during the analysis. |

## Reporting for specific materials, systems and methods

We require information from authors about some types of materials, experimental systems and methods used in many studies. Here, indicate whether each material, system or method listed is relevant to your study. If you are not sure if a list item applies to your research, read the appropriate section before selecting a response.

### Materials & experimental systems

| n/a | Involved in the study |
|---|---|
| ☒ | ☐ Antibodies |
| ☐ | ☒ Eukaryotic cell lines |
| ☒ | ☐ Palaeontology and archaeology |
| ☒ | ☐ Animals and other organisms |
| ☒ | ☐ Clinical data |
| ☒ | ☐ Dual use research of concern |
| ☒ | ☐ Plants |

### Methods

| n/a | Involved in the study |
|---|---|
| ☒ | ☐ ChIP-seq |
| ☒ | ☐ Flow cytometry |
| ☒ | ☐ MRI-based neuroimaging |

## Eukaryotic cell lines

Policy information about cell lines and Sex and Gender in Research

| | |
|---|---|
| Cell line source(s) | SF9, Oxford Expression Technologies Ltd, Cat No. 600100 |
| Authentication | Not authenticated |
| Mycoplasma contamination | Cell stocks tested for mycoplasma contamination monthly. |
| Commonly misidentified lines (See ICLAC register) | N/A |

## Plants

| | |
|---|---|
| Seed stocks | n/a |
| Novel plant genotypes | n/a |
| Authentication | n/a |

