## [Peer Review File · Nature Structural & Molecular Biology]

Phosphorylation-dependent tuning of mRNA deadenylation rates

Corresponding Author: Dr Lori Passmore

Version 0:

Decision Letter:

25th Nov 2024

Dear Dr Passmore,

Thank you again for submitting your manuscript "Phosphorylation-dependent tuning of mRNA deadenylation rates". I apologize for the delay in sharing our decision with you. We now have comments (below) from the 3 reviewers who evaluated your paper. In light of those reports, we remain interested in your study and would like to see your response to the comments of the referees in the form of a revised manuscript.

You will see that the reviewers found the dataset of interest but also raised important concerns about the strength of the biochemical and biophysical analyses, which are essential to quantitatively support your model. We have discussed the reviewers' feedback in detail within our team. The referees' concerns are significant and in our view would need to be addressed thoroughly experimentally; reconsideration of the study for this journal and re-engagement of referees will depend on the strength of these revisions.

To guide the scope of the revisions, we list below a prioritized set of referee points that should be addressed in the revision, which we hope will be helpful to you.

In our view, the reviewers' concerns about the mechanism and functional aspects need to be strengthened and addressed in full (all major comments from Rev#1; Rev#3 points #1, 2, 3, 4, 6, 8, 9, 11) as the reviewers' concerns question the multivalent mechanism, and Rev#3's concerns about the NMR data (#5, #7, #10) should also be addressed robustly. Please also address the other technical questions, minor requests, reviewer points about strengthening existing data and additional discussion or text edits.

We are committed to providing a fair and constructive peer-review process. Please do not hesitate to contact us if there are specific requests from the reviewers that you believe are technically impossible or unlikely to yield a meaningful outcome. Please be sure to address and respond to all concerns of the referees in full in a point-by-point response and highlight all changes in the revised manuscript text file.

We appreciate the requested revisions are extensive; our standard revision period is six months. If you cannot send it within this time, please let us know. We will be happy to consider your revision as long as nothing similar has been accepted for publication at NSMB or published elsewhere. Should your manuscript be substantially delayed without notifying us in advance and your article is eventually published, the received date would be that of the revised, not the original, version.

Reporting Summary:
<https://www.nature.com/documents/nr-reporting-summary.pdf>

Please note that all key data shown in the main figures as cropped gels or blots should be presented in uncropped form, with molecular weight markers. These data can be aggregated into a single supplementary figure. While these data can be displayed in a relatively informal style, they must refer back to the relevant figures. These data should be submitted with the last revision, prior to acceptance, but you may want to start putting it together at this point.

We require deposition of coordinates (and, in the case of crystal structures, structure factors) into the Protein Data Bank with the designation of immediate release upon publication (HPUB). Electron microscopy-derived density maps and coordinate data must be deposited in EMDB and released upon publication. Deposition and immediate release of NMR chemical shift assignments are highly encouraged. Deposition of deep sequencing and microarray data is mandatory, and the datasets must be released prior to or upon publication. To avoid delays in publication, dataset accession numbers must be supplied with the final accepted manuscript and appropriate release dates must be indicated at the galley proof stage. Please find the complete NRG policies on data availability at <http://www.nature.com/authors/policies/availability.html>.

Link Redacted

We look forward to seeing the revised manuscript and thank you for the opportunity to review your work. Thank you again for considering our journal for your work.

Sincerely,

Melina

Melina Casadio, PhD
Consulting Editor, Nature Structural & Molecular Biology
Senior Editor, Nature Cell Biology
ORCID ID: <https://orcid.org/0000-0003-2389-2243>

Referee expertise:

Referee #1: RNA decay pathways, RNA metabolism

Referee #2: NMR, RNA

Referee #3: NMR, RNA

Reviewers' Comments:

Reviewer #1 (Remarks to the Author):

This is a very elegant study dissecting the role of the intrinsically disordered region (IDR) of the yeast Pumilio protein Puf3 in mediating interactions with the Ccr4-Not deadenylase complex, which allows targeting of Puf3-bound mRNAs for rapid deadenylation. The authors make use of NMR spectroscopy, crosslinking mass spectrometry and in vitro deadenylation assays to unravel detailed structural and functional aspects of the Puf3:Ccr4-Not interaction. The main result of the study is that the Puf3 IDR uses multiple distant residues to contact the Ccr4-Not complex, and that these interactions together have an additive effect in mediating the function of Puf3. Moreover, the authors demonstrate that progressive phosphorylation of the Puf3 IDR reduces the ability of Puf3 to interact with the Ccr4-Not complex and stimulate deadenylation. In the last part of their study, the authors extend this concept by showing progressive binding of the human PUM1 IDR to the human CCR4-NOT complex, and providing evidence that phosphomimetic mutations in the IDR of TTP progressively reduce its ability to target an mRNA for deadenylation via CCR4-NOT.

While previous studies have focused on identifying short linear motifs (SLIMs) within IDRs of RNA-binding proteins that mediate interactions with the CCR4-NOT complex, this study goes two steps further by i) structurally elucidating the multivalency of such IDR interactions, and ii) providing evidence that progressive phosphorylation interferes with these interactions, offering a mechanistic explanation for the observed regulation of RNA-binding proteins by phosphorylation. I am quite excited about the results presented here and have only two major comments that could strengthen the study. Several minor comments are aimed at improving representation of the results.

Major comments:

Fig. 2B: The authors should also measure the K_d for a full-length Puf3 mutant lacking only the DFW motif (Δ 217-232:GSN). This would give a better idea of the contribution of this motif to overall binding.

Given their finding that Puf3 appears to mainly interact with the Not9 subunit of the complex, the authors should test how potently Puf3 can induce mRNA deadenylation and/or decay when Not9 is deleted from cells. Alternatively, this question could be addressed through the author's in vitro deadenylation assay by using a Ccr4-Not complex lacking Not9. This would clarify if/how important Not9 is in mediating the activity of Puf3.

Minor comments:

Fig. 1C and D; Extended Data Fig. 2A and 3A: grey and blue does not give much contrast. I suggest that the authors use a different color-code to better distinguish the conditions.

Line 85: The authors should specify which subunits are contained in what they refer to as the „full-length 0.5-MDa unlabeled Ccr4-Not complex“.

EV Data Fig. 5: This is a very nice way to quantify deadenylation kinetics - it should also be applied to the Puf3 mutants in Fig. 1B.

Fig. 2A and Extended Data Fig. 5C: What is the rationale for mutating the single Puf3 residues to asparagin (and not to other amino acids)?

Does the L43N mutation have an effect on the deadenylation rate? Fig. 2A suggest that L43 does not contribute. From Extended Data Fig. 5C, this is hard to judge since the single W229N mutant is not depicted in this graph.

In general, it would be helpful to compare the deadenylation kinetics of Puf3 mutants in summary plots as is done for TTP mutants in Extended Data Fig. 15B.

Line 113: "... extensive regions of the Puf3 IDR, including the DFW motif, are required for interaction with Ccr4-Not." This sentence is misleading as none of the regions alone are fully required for binding to the complex. Rather, multiple regions appear to contribute to the interaction. The authors should rephrase this.

Fig. 3B: Authors should specify in the text or legend which residues were mutated in "Not9 mutant" and "Not1,3,9 mutant". This is only visible if one looks very carefully at Extended Data Fig. 8A.

Fig. 4D: Is this the same data as the bottom row of Extended Data Fig. 10C?

Reviewer #2 (Remarks to the Author):

Stowell et al investigate the molecular mechanism of how sequence-specific RNA adaptor proteins promote transcript specific deadenylation by the conserved Ccr4/Not complex, the first step in many eukaryotic mRNA decay pathways. Prior studies have established Pumilio and FBF (Puf) protein family members as paradigmatic examples of an RNA adaptor protein for the Ccr4/Not complex. Puf family members and related adaptors contain a structured, sequence-specific RNA binding domain (RBD) and one or more intrinsically disordered regions (IDRs) that interact with the 3' untranslated region of

a mRNA and the Ccr4/Not complex respectively thereby tethering the deadenylase to the transcript to promote gene-specific mRNA decay.

While fragmented structural data supported by functional studies document how SLiMs embedded in IDRs of various adaptor proteins interact with specific domains of the Ccr4/Not complex, a comprehensive picture of the interactions of intact full-length adaptors with Ccr4/Not is lacking. This is because IDRs are extremely challenging to study by conventional structural methods such as X-ray crystallography or cryoEM. Stowell et al employ a tour de force combination of quantitative biochemistry, NMR, cross-linking MS and machine learning based structure prediction, to provide a comprehensive view on how Puf3 and related adaptor proteins regulate deadenylation by Ccr4/Not. First, using NMR chemical shift and cross-peak intensity perturbation analyses, they discover multiple SLiMs in the IDR of Puf3 that engage Ccr4/Not, including a SLiM that contains the amino acids D, F and W (DFW motif) found in all *Schizosaccharomyces* species. Second, they show using mutagenesis and quantitative deadenylation assays that multiple motifs in the Puf3 IDR identified by NMR are required to promote removal of the poly(A) tail by Ccr4/Not. Third, using CLMS the authors map interactions between different subunits of the CCR4/Not complex with the IDR of Puf3. This information is corroborated using machine-learning based structure prediction and kinetic analyses of mutants lining the cognate binding surfaces on different Ccr4/Not subunits. The results are consistent with the notion that multiple motifs in the IDR of Puf3 contact multiple different subunits of the Ccr4/Not complex. Fourth, the authors show that phosphorylation of the IDR of Puf3 occurs near a conserved DFW containing SLiM prompting them to study the relationship between phosphorylation of Puf3 and Ccr4/Not activity. They find that graded phosphorylation of the Puf3 IDR reduces its interaction with Ccr4/Not complex, functioning as a rheostat to tune deadenylation activity.

This manuscript is suitable for the broad readership of NSMB because it provides a playbook for how to study interactions of IDRs of effector proteins with large multi-subunit molecular machines and suggests that one function of IDRs is to impart tunability of multivalent interaction for regulation of activity. The manuscript is well written, the biochemical experiments are well controlled, and the NMR experiments on the Puf3 IDR alone and in complex with the 0.5 MDa Ccr4/Not complex were conducted with virtuosity.

Accordingly, publication of the manuscript is recommended after the following minor comments are addressed:

1. ED Figure 1A is called in the first and second paragraph of the manuscript but this figure has only one panel, which is not labeled A.
2. Related, ms p3 line 96 it is stated that Trp motifs in GW182 are important for interaction with Not1. This should be indicated on ED Fig 1.
3. I count around 12 cross-peaks in phosphoserine/threonine region of the HSQC but the authors reported they 'observed 9 new resonances' are the author cross peaks in the region of 8.6-8.8pp noise or minor conformations in slow exchange?
4. The rate of deadenylation in the absence of Puf3 would be informative in order to show the extent to which mutations ablate the stimulation of Ccr4-Not activity by Puf3. Please include in Fig 2A; 3B, 4F and 5A-C.
5. Fig. 5B and page 6, line 228: This experiment shows that the TTP N-terminal IDR is largely dispensable for stimulation of deadenylation. Previous work looking at the binding between TTP and CNOT9 showed that deletion of the TTP C-terminal IDR led to much more significant decrease in binding than did deletion of the TTP N-terminal IDR (DOI: 10.1016/j.jmb.2017.12.018). This is likely due to meaningful experimental differences between these results (isolated CNOT9 vs. full CCR4-NOT complex; binding vs. deadenylase activity), but it is still worth noting this discrepancy in the manuscript text.
6. Fig. 5C: The phosphomimic mutations in TTP appear to perturb the activation of CCR4-NOT to a lesser extent than does phosphorylation of Puf3 (Fig. 4F). Is this due to lesser activation by wild-type TTP than by unphosphorylated Puf3 (leading to a smaller dynamic range)? Is it due to the limitations of phosphomimic mutations? Or does this reflect something deeper about the differences in the biological role of these two RNA adaptor proteins? The answers to these questions are worth noting in the manuscript text.
7. The caption for ED Fig 6 D is confusing. Please state the complex between Not9 and Not1 is depicted. '...AlphaFold2 prediction of *S. pombe* Not9 and Not1' suggest separate molecules might be displayed.
8. For ED Fig 7B in the PAE plots it looks like there may be trade-offs in accuracy of the predicted models Not9/Not1 with Puf3 containing one or two omega pockets are engaged. Do the authors have an interpretation of this observation?
9. The authors should state why the MBP tag for Puf3 was not removed for biochemical studies.

Reviewer #3 (Remarks to the Author):

This work analyses molecular interactions between the deadenylation complex Ccr4-Not and RNA adaptor proteins, such as Puf3. The interactions are characterized using biochemical assays (deadenylation assays), NMR spectroscopy and mass spectrometry-detected cross-links. The study aims at being an extensive biophysical and biochemical analysis of individual interactions and their functional significance. However, I am confused both by the presentation of the data, their analysis and

the conclusions drawn from them. In details:

1. In the absence of phosphorylation, activity assays are presented in Figure 1, Extended Data Figure 5 and 8. A relative quantification is done in Extended Data Figure 5 but not in the other Figures. For the mutants presented in ED Fig. 5, it is stated that they affect deadenylation efficiency much less than the deletion mutants in Fig. 1. How much less? Why are the data in Fig. 1 not quantified relatively to each other? Even for the 258 mutant, where the rate of deadenylation is probably too slow to be properly quantified, an upper limit could be provided.
2. The effect of the individual mutations of ED Fig. 5 is small and does not reach that of the 258 mutant even when combining all mutations. Nevertheless, it is argued that the effect of the mutations is additive. Can the authors specify what they mean by additive? Is this a quantitative or a qualitative statement? I am not sure that the word additive is appropriate, as I do not see that this "additivity" is in quantitative agreement with the data of ED Fig. 5. The additivity is nicely apparent in the selection of data given in Fig. 2. However, from ED Fig. 5C it appears that F65N has no additional effect when combined with L43N and W229N, while it has an effect by itself (ED Fig. 5D). On the other hand, an effect seems to exist for F65N in combination with L43N and W229N in Fig. 2. Similarly, Fig. 2 shows no effect for L43N in combination with W229N, while an effect is seen in ED Fig. 5C for L43N added to the combination W229N and F65N. Either the data of the two figures are inconsistent with each other or the effects are not additive.
3. The first half of the paper suggests that the affinity of the N-terminal disordered region of Puf3 for Ccr4-Not complex is correlated to the efficiency of deadenylation. While this is qualitatively supported by the data, it is not supported quantitatively. To obtain a quantitative support, the same mutants for which binding assays are presented in Figure 2 should also be tested in adenylation assays and vice-versa. For example, the affinities of Puf3109-258 and Puf3158-258 for Ccr4-Not differ by a factor of 10. How do the deadenylation activities of D109 and D158 compare with each other?
4. Why were affinities of Fig. 2 measured only with the N-terminal part? The disordered N-terminal region may have interaction in the context of the entire Puf3 protein that change its affinity for Ccr4-Not. The affinities of the N-terminal-only constructs should be compared with those of D109, D158 and D258 constructs.
5. The NMR data should be analysed more carefully. A quick inspection of the spectra reveals that line shapes differ between apo and holo spectra. Instead of intensities, a more appropriate quantification would include FUDA fits of the peaks. No explanation is given about how the intensities were extracted. Was a fitting program used? How were spectra processed? All these factors influence intensity values. The effect in the region 10-80 are not continuous: is the assumption here that only individual amino acids contribute to the interaction with Ccr4-Not, while the rest are flexible? The data at different Ccr4-Not concentrations are not completely consistent. Any ideas why this is the case? In addition, the region between 210-260 seems to be less affected at higher concentrations of Ccr4-Not, which makes little sense. A proper fitting of the data, considering overlaps and excluding peaks that are too overlapped to be quantified, even with a proper line-fitting program, might shed light on these inconsistencies.
6. A discussion is needed about the exchange regime of the Puf3 N-terminal region with Ccr4-Not. Clearly, as the peaks disappear at sub-stoichiometric concentrations of Ccr4-Not, the exchange rate between bound and free forms of Puf3 is not slow. Have the authors attempted a more quantitative analysis of the line-broadening in dependence of the concentration? For example, the smaller line-broadening observed for the 10-80 region may indicate a lower affinity of this part of the protein for Ccr4-Not (i.e. a different exchange regime) or a different "rigidity" of this region when bound. Again, to prove the "additivity" argument, the contribution of the 10-80 region to the binding of Ccr4-Not should be investigated more thoroughly by quantitatively comparing the NMR data to the contribution of this region to affinity, as determined by other methods. Another possibility is that the 10-80 region does not have any direct interaction with Ccr4-Not but folds upon the 158-260 region (when this is interacting with Ccr4-Not) and stabilizes its interaction with Ccr4-Not. In this case, the presented conclusion about "multivalency" would be wrong.
7. The cross-link data do not seem to match well with the NMR data, but this fact is neither addressed nor discussed. Most of the cross-links seem to occur in the central region of Puf3, which is not consistently affected in NMR spectra. It is well known that cross-links may reveal also transient interactions and may overweight them with respect to more stable interactions. Is this what is happening here? Why do we not see any cross-links from the 210-260 region?
8. The observed cross-links are also not in agreement with the models presented in ED Fig. 6. In this figure only the region 210-260 of Puf3 interacts with Not9-Not1. Where is the region upstream of it that forms most cross-links?
9. In ED Fig. 6 the prediction quality seems to be higher for the individual residues that interact with the aromatic pocket(s) than for the residues that interact with the peptide binding pocket, whose prediction score is rather poor. I am not sure what to make of it in terms of the overall trustability of these models. Probably quite poor overall.
10. The phosphorylation bit is valuable, but again I do not understand the experimental design. Why were NMR experiments measuring relaxation performed only for Puf3109-258, if we know that the residues 10-80 also bind Ccr4-Not? In the paper, this is justified by the need to reduce heterogeneity, but I do not understand which heterogeneity is meant here. At least, it should be checked that relaxation parameters of the 109-258 stretch w/o phosphorylation do not change between the 1-258 and 109-258 constructs.
11. The authors seem to assume that individual disordered stretches in the context of a long, disordered contrasts do not interact with each other. This is evident from the experimental design (see previous point) and from the last paragraph (before Discussion) supposedly presenting proofs of a general multivalency recognition mechanism. This assumption has been demonstrated wrong in many papers and is a rather simplistic way to view disordered proteins. These days we know that disordered-disordered interactions are functional. When they observe an increasing functional impairment upon deletion of longer IDR stretches, the authors conclude that this data indicates the presence of multivalent interactions. This is not necessarily the case. Disordered regions have internal interactions and residual structures and are not equal to bids on a chain. Individual disordered stretches of a long, disordered domain cannot be treated as independent from each other. Thus, the experimental data presented in the last paragraph do not support the conclusion of a "universal" multivalency mechanism (see also point 6). Multivalency can be at play here, but the data do not demonstrate it robustly enough and further experiments/analysis are needed.

12. To ED Figure 15, in the example of TTP, the deletion of the N-term and C-term are all but additive! There is cooperativity between the N-term and the C-term, which goes beyond multivalency.

Other points:

In the introduction, it is confusing to use many different names for the same protein. If this is done, then it should be explicitly stated to which organism each name applies. The way it stands is jargon for experts. Many abbreviations are not spelled out.

In conclusion, although the topic of the paper is of interest, the biochemical and biophysical analysis lacks the consistency and rigor that I would expect for a manuscript in NSMB. The concept of multivalency is trivial if not substantiated by a quantitative analysis. This work lacks a rigorous quantification of the contributions to binding of the different regions of Puf3, the multiple, interdisciplinary assays are not applied to a consistent set of constructs, the NMR data are superficially analysed, the cross-link data are inconsistent with all other data and there is no information of which part of Puf3 interacts with which part of Ccr4-Not. In addition, the authors do not consider that a long, disordered region of 258 residues may not be a series of bids on a chain but may entertain disorder-to-disorder interactions. These interactions underline the necessity of using the full-length construct or at least of running the appropriate controls. In addition, data interpretation need to be revised.

Version 1:

Decision Letter:

Our ref: NSMB-A49918A

30th Jul 2025

Dear Dr. Passmore,

Thank you for submitting your revised manuscript "Phosphorylation-dependent tuning of mRNA deadenylation rates" (NSMB-A49918A). It has now been seen by the original referees and Rev#4 (replacing Rev#3 who was unavailable; Rev#2 also helped us assess the revisions made in response to Rev#3) and their comments are below. The reviewers find that the paper has improved in revision, and therefore we'll be happy in principle to publish it in Nature Structural & Molecular Biology, pending minor revisions to satisfy the referees' final requests and to comply with our editorial and formatting guidelines.

We are now performing detailed checks on your paper and will send you a checklist detailing our editorial and formatting requirements in about 1-2 weeks. Please do not upload the final materials and make any revisions until you receive this additional information from us.

To facilitate our work at this stage, it is important that we have a copy of the main text as a word file. If you could please send along a word version of this file as soon as possible, we would greatly appreciate it; please make sure to copy the NSMB account (cc'ed above).

Thank you again for your interest in Nature Structural & Molecular Biology. Please do not hesitate to contact me if you have any questions.

Sincerely,

Melina Casadio, PhD
Locum Chief Editor, Nature Structural & Molecular Biology
ORCID ID: <https://orcid.org/0000-0003-2389-2243>

Reviewer #1 (Remarks to the Author):

The authors have addressed all my concerns in this revision and substantially strengthened their manuscript. This is an important study for the RNA decay field and for IRD research, and I am looking forward to seeing it published as soon as possible.

Georg Stoecklin

Reviewer #2 (Remarks to the Author):

The authors have addressed my questions and comments in their revised manuscript. It's an outstanding study that uses a wide range of techniques to biochemically characterize an extremely challenging system. The work goes beyond just the area of deadenylation/mRNA decay, given its quantitative dissection of the multivalent interaction and the "rheostat" model

of tuning via phosphorylation.

Minor suggestion: In the legend for Fig 4e, the authors should indicate that red and black bars correspond to phosphorylated and unphosphorylated forms of Puf3, respectively.

Additionally, the authors have addressed the comments and questions of Reviewer 3. Most notably, they have clarified the rationale for the constructs studied by NMR, fluorescence anisotropy and deadenylation assays and have added additional protein interaction (pull-down) experiments to address the correlation between binding and the ability of the Puf3 IDR to stimulate deadenylation activity. Additionally, they have fortified their NMR analysis with measurements of ¹⁵N R2 relaxation rates for the IDR of Puf3 to monitor changes in backbone mobility as a function of binding to Ccr4/Not and phosphorylation. Lastly, they have changed their interpretation of 'additivity' of IDR regions, taking into account comments on cooperative interactions as evidenced by mutational data.

Overall, this manuscript is now suitable for publication in NSMB.

Reviewer #4 (Remarks to the Author):

The authors have addressed the criticisms raised by the reviewers in a constructive manner. In particular, the new "quantifications" and the analysis of the correlation between deadenylase activity and molecular interactions strengthen the conclusions and add depth to the study. The manuscript in its current form represents a significant contribution to the field and is suitable for publication.

There is one minor inconsistency in the response to point 3, where Extended Data 3 is referenced, though it appears the correct figure is Extended Data 4 ("There are some deviations but these are likely due to the limitations mentioned above and now discussed in the legend for Extended Data Fig. 3")

Version 2:

Decision Letter:

8th Sep 2025

Dear Dr. Passmore,

We are now happy to accept your revised paper "Phosphorylation-dependent tuning of mRNA deadenylation rates" for publication as a Article in Nature Structural & Molecular Biology.

Your paper will be published online soon after we receive proof corrections and will appear in print in the next available issue. You can find out your date of online publication by contacting the production team shortly after sending your proof corrections.

Authors may need to take specific actions to achieve compliance with funder and institutional open access mandates. If your research is supported by a funder that requires immediate open access (e.g. according to <https://www.springernature.com/gp/open-science/plan-s-compliance> Plan S principles or the <https://www.springernature.com/gp/open-science/us-federal-agency-compliance> NIH public access policy) then you should select the gold OA route, and we will direct you to the compliant route where possible. Because authors warrant under our subscription licensing terms that they haven't committed to licensing any version of their article under a licence inconsistent with the terms of our agreement – including the applicable embargo period – publication under the subscription model isn't suitable for authors whose funders require no embargo.

Sincerely,

Melina Casadio, PhD
Consulting Senior Editor, Nature Structural & Molecular Biology
ORCID ID: <https://orcid.org/0000-0003-2389-2243>

We thank the Reviewers for their careful analysis of our manuscript. Our revised manuscript is much improved. We have reanalysed and collected new NMR data, performed a new AlphaFold screen and performed new binding and deadenylation assays. Major changes in the manuscript file are marked with red text.

Reviewer #1 (Remarks to the Author):

This is a very elegant study dissecting the role of the intrinsically disordered region (IDR) of the yeast Pumilio protein Puf3 in mediating interactions with the Ccr4-Not deadenylase complex, which allows targeting of Puf3-bound mRNAs for rapid deadenylation. The authors make use of NMR spectroscopy, crosslinking mass spectrometry and in vitro deadenylation assays to unravel detailed structural and functional aspects of the Puf3:Ccr4-Not interaction. The main result of the study is that the Puf3 IDR uses multiple distant residues to contact the Ccr4-Not complex, and that these interactions together have an additive effect in mediating the function of Puf3. Moreover, the authors demonstrate that progressive phosphorylation of the Puf3 IDR reduces the ability of Puf3 to interact with the Ccr4-Not complex and stimulate deadenylation. In the last part of their study, the authors extend this concept by showing progressive binding of the human PUM1 IDR to the human CCR4-NOT complex, and providing evidence that phosphomimetic mutations in the IDR of TTP progressively reduce its ability to target an mRNA for deadenylation via CCR4-NOT.

While previous studies have focused on identifying short linear motifs (SLIMs) within IDRs of RNA-binding proteins that mediate interactions with the CCR4-NOT complex, this study goes two steps further by i) structurally elucidating the multivalency of such IDR interactions, and ii) providing evidence that progressive phosphorylation interferes with these interactions, offering a mechanistic explanation for the observed regulation of RNA-binding proteins by phosphorylation. I am quite excited about the results presented here and have only two major comments that could strengthen the study. Several minor comments are aimed at improving representation of the results.

We thank the reviewer for their constructive comments.

Major comments:

1. Fig. 2B: The authors should also measure the K_d for a full-length Puf3 mutant lacking only the DFW motif ($\Delta 217-232$:GSN). This would give a better idea of the contribution of this motif to overall binding.

We have now measured the apparent dissociation constant (K_{Dapp}) for the $\Delta 217-232$:GSN Puf3 construct. We also included $\Delta 53-69$:GSN and the combined $\Delta 53-69,217-232$:GSN Puf3 mutant since this more N-terminal region shows intensity and backbone relaxation changes on Ccr4-Not binding (see also comments to Reviewer 3, point 5). These new binding data show that $\Delta 217-232$:GSN has ~2-fold reduced binding affinity for Ccr4-Not while $\Delta 53-69$:GSN Puf3 has a similar binding affinity as WT Puf3 (**Extended Data Fig. 4f**). Interestingly, the double mutant had a larger effect – we were not able to measure the K_{Dapp} for the double mutant as it did not reach saturation.

We have also now performed deadenylation assays for these mutants: The single region substitution mutants show an ~2-fold reduced ability to stimulate deadenylation by Ccr4-Not, while the double substitution mutant has ~3-fold reduced activity (**Fig. 2a**).

Overall, these data further support a model where lower affinity binding is generally correlated with lower activity in deadenylation assays, and multiple distinct regions of Puf3 contribute to binding in a multivalent manner. There is not necessarily a one:one correlation between deadenylation activity and binding affinity – in this case the double substitution mutant has a larger effect on binding than on deadenylation activity and $\Delta 53-69$:GSN Puf3 has a stronger effect on deadenylation than binding. The reasons for this may be due to our assay setup and are discussed further in response to Reviewer 3, point 3.

In summary, substitution of the DFW motif (residues 217-232) results in ~2-fold reduction in binding and activity, whereas additional substitution of residues 53-69 substantially reduces binding, suggesting that multiple (longer) regions of the Puf3 IDR contribute, possibly in a cooperative manner. We have modified the text accordingly.

2. Given their finding that Puf3 appears to mainly interact with the Not9 subunit of the complex, the authors should test how potently Puf3 can induce mRNA deadenylation and/or decay when Not9 is deleted from cells. Alternatively, this question could be addressed through the author's in vitro deadenylation assay by using a Ccr4-Not complex lacking Not9. This would clarify if/how important Not9 is in mediating the activity of Puf3.

This is an excellent suggestion. We produced a Ccr4-Not complex lacking Not9 using the same method as with the wild-type complex, by baculovirus mediated co-expression of the remaining subunits. We were able to purify this complex lacking Not9 as confirmed by SDS-PAGE, but it became apparent that it was not monodisperse in solution: SEC-MALS analysis shows that the complex lacking Not9 is a mixture of higher order oligomers (**Fig. R1a**). Deadenylation by Ccr4-Not^{ΔNot9} is not substantially compromised in the absence of Puf3. Any comparison with the activity of the WT complex would therefore need to be interpreted with caution. Nevertheless, we tested the ability of Puf3 to accelerate the deadenylation activity of this complex and found that the activity was reduced by ~5-fold (**Fig. R1b**). Therefore, complex lacking Not9 has a partial defect in Puf3-mediated stimulation of deadenylation, consistent with binding sites for Puf3 on Not9 as well as on other Ccr4-Not subunits. Given the limitations of this experiment, we have not included these data in the final manuscript.

Figure R1: Purification and assay of Ccr4-Not lacking the Not9 subunit. (a) Size exclusion chromatography coupled to mass spectrometry (SEC-MALS) of purified Ccr4-Not (green), Ccr4-Not lacking Not9 (red) and a BSA control (blue). Ccr4-Not lacking Not9 shows aggregation. (b) Deadenylation assays of the Puf3-substrate mRNA with a 30-nt poly(A) tail using Ccr4-Not^{ΔNot9} with and without Puf3. With Puf3, Ccr4-Not^{ΔNot9} is approximately 5-fold slower compared to wild-type Ccr4-Not (compare to Fig. 1b).

Given the problems with monodispersity of Ccr4-Not lacking Not9, we did not pursue cellular studies. Still, a mutation in the peptide binding groove of Not9 reduces the ability of Puf3 to activate deadenylation (**Fig. 3d**). Thus, a combination of binding sites on Ccr4-Not that interact with several regions in the 258 residue IDR is important (e.g. as discussed above for the two regions where we made GSN substitutions, **Extended Data Fig. 4f**). We have made this clearer in the manuscript.

Minor comments:

3. Fig. 1C and D; Extended Data Fig. 2A and 3A: grey and blue does not give much contrast. I suggest that the authors use a different color-code to better distinguish the conditions.

We have now changed the colours in **Fig. 1** and in the accompanying Extended Data figures, either to completely different colours or to different shades. We hope that this facilitates interpretation.

4. Line 85: The authors should specify which subunits are contained in what they refer to as the „full-length 0.5-MDa unlabeled Ccr4-Not complex“.

We now list the subunits in the text: “...the full-length 0.5-MDa unlabeled Ccr4-Not complex (Not1, Not2, Not3, Not4, Not9, Ccr4 and Caf1)...”

5. EV Data Fig. 5: This is a very nice way to quantify deadenylation kinetics - it should also be applied to the Puf3 mutants in Fig. 1B.

We have now included quantitation of deadenylation assays in **Fig. 1B**. For transparency, we now plot the activity as half-lives ($t_{1/2}$) instead of relative rates so all figures are directly comparable.

6. Fig. 2A and Extended Data Fig. 5C: What is the rationale for mutating the single Puf3 residues to asparagin (and not to other amino acids)?

All residues that we mutated were aromatic or hydrophobic; we wanted to mutate these without changing the overall charge on the IDR. Thus, we mutated them to asparagine.

7. Does the L43N mutation have an effect on the deadenylation rate? Fig. 2A suggest that L43 does not contribute. From Extended Data Fig. 5C, this is hard to judge since the single W229N mutant is not depicted in this graph.

We have included data for all individual mutants (now in **Extended Data Fig. 3c**). L43N has a minor effect on deadenylation rate.

8. In general, it would be helpful to compare the deadenylation kinetics of Puf3 mutants in summary plots as is done for TTP mutants in Extended Data Fig. 15B.

We have added the summary plots for assays in **Extended Data Fig. 1d** and for assays with human proteins (now in **Extended Data Fig. 9-10**).

9. Line 113: "... extensive regions of the Puf3 IDR, including the DFW motif, are required for interaction with Ccr4-Not." This sentence is misleading as none of the regions alone are fully required for binding to the complex. Rather, multiple regions appear to contribute to the interaction. The authors should rephrase this.

We have rephrased this to: *"Overall, these data support a model where extensive regions of the Puf3 IDR contribute to the interaction with Ccr4-Not."*

10. Fig. 3B: Authors should specify in the text or legend which residues were mutated in "Not9 mutant" and "Not1,3,9 mutant". This is only visible if one looks very carefully at Extended Data Fig. 8A.

We have added a note to the legend of Fig. 3d that the mutations are listed in Extended Data Fig. 6a.

11. Fig. 4D: Is this the same data as the bottom row of Extended Data Fig. 10C?

Yes, this was the same set of data. We have now removed it from the Extended Data Figure for clarity.

Reviewer #2 (Remarks to the Author):

Stowell et al investigate the molecular mechanism of how sequence-specific RNA adaptor proteins promote transcript specific deadenylation by the conserved Ccr4/Not complex, the first step in many eukaryotic mRNA decay pathways. Prior studies have established Pumellio and FBF (Puf) protein family members as paradigmatic examples of an RNA adaptor protein for the Ccr4/Not complex. Puf family members and related adaptors contain a structured, sequence-specific RNA binding domain (RBD) and one or more intrinsically disordered regions (IDRs) that interact with the 3' untranslated region of a mRNA and the Ccr4/Not complex respectively thereby tethering the deadenylase to the transcript to promote gene-specific mRNA decay.

While fragmented structural data supported by functional studies document how SLiMs embedded in IDRs of various adaptor proteins interact with specific domains of the Ccr4/Not complex, a comprehensive picture of the interactions of intact full-length adaptors with Ccr4/Not is lacking. This is because IDRs are extremely challenging to study by conventional structural methods such as X-ray crystallography or cryoEM. Stowell et al employ a tour de force combination of quantitative biochemistry, NMR, cross-linking MS and machine learning based structure prediction, to provide a comprehensive view on how Puf3 and related adaptor proteins regulate deadenylation by Ccr4/Not. First, using NMR chemical shift and cross-peak intensity perturbation analyses, they discover multiple SLiMs in the IDR of Puf3 that engage Ccr4/Not, including a SLiM that contains the amino acids D, F and W (DFW motif) found in all *Schizosaccharomyces* species. Second, they show using mutagenesis and quantitative deadenylation assays that multiple motifs in the Puf3 IDR identified by NMR are required to promote removal of the poly(A) tail by Ccr4/Not. Third, using CLMS the authors map interactions between different subunits of the CCR4/Not complex with the IDR of Puf3. This information is corroborated using machine-learning based structure prediction and kinetic analyses of mutants lining the cognate binding surfaces on different Ccr4/Not subunits. The results are consistent with the notion that multiple motifs in the IDR of Puf3 contact multiple different subunits of the Ccr4/Not complex. Fourth, the authors show that phosphorylation of the IDR of Puf3 occurs near a conserved DFW containing SLiM prompting them to study the relationship between phosphorylation of Puf3 and Ccr4/Not activity. They find that graded phosphorylation of the Puf3 IDR reduces its interaction with Ccr4/Not complex, functioning as a rheostat to tune deadenylation activity.

This manuscript is suitable for the broad readership of NSMB because it provides a playbook for how to study interactions of IDRs of effector proteins with large multi-subunit molecular machines and suggests that one function of IDRs is to impart tunability of multivalent interaction for regulation of activity. The manuscript is well written, the biochemical experiments are well controlled, and the NMR experiments on the Puf3 IDR alone and in complex with the 0.5 MDa Ccr4/Not complex were conducted with virtuosity.

We appreciate the Reviewer's comments.

Accordingly, publication of the manuscript is recommended after the following minor comments are addressed:

1. ED Figure 1A is called in the first and second paragraph of the manuscript but this figure has only one panel, which is not labeled A.

Thank you – we have amended this.

2. Related, ms p3 line 96 it is stated that Trp motifs in GW182 are important for interaction with Not1. This should be indicated on ED Fig 1.

We have now indicated GW182 binding to the C-terminal region of NOT1 on this figure.

3. I count around 12 cross-peaks in phosphoserine/threonine region of the HSQC but the authors reported they 'observed 9 new resonances' are the author cross peaks in the region of 8.6-8.8pp noise or minor conformations in slow exchange?

These extra peaks represent additional unassigned phosphorylated residues, or split peaks from assigned residues where there are neighbouring phosphorylations in a fraction of the sample, e.g. pS156, **Extended Data Fig. 7c**. We have updated the manuscript accordingly.

4. The rate of deadenylation in the absence of Puf3 would be informative in order to show the extent to which mutations ablate the stimulation of Ccr4-Not activity by Puf3. Please include in Fig 2A; 3B, 4F and 5A-C.

We added deadenylation rates for a construct that contains only the PUM domain (**Fig. 1b**), which is similar to the activity of Ccr4-Not in the absence of Puf3 (**Extended Data Fig. 1c**). Due to the large difference between WT Puf3 and the PUM domain alone (~50-fold), it is not helpful to add this directly to all of the figures. Instead, we now report $t_{1/2}$, instead of relative deadenylation activity so that the experiments in different figures are directly comparable.

5. Fig. 5B and page 6, line 228: This experiment shows that the TTP N-terminal IDR is largely dispensable for stimulation of deadenylation. Previous work looking at the binding between TTP and CNOT9 showed that deletion of the TTP C-terminal IDR led to much more significant decrease in binding than did deletion of the TTP N-terminal IDR (DOI: 10.1016/j.jmb.2017.12.018). This is likely due to meaningful experimental differences between these results (isolated CNOT9 vs. full CCR4-NOT complex; binding vs. deadenylase activity), but it is still worth noting this discrepancy in the manuscript text.

We had referenced this study in our paper (reference 24). That study showed that the N-terminal IDR region contributed slightly more to binding isolated NOT9 in vitro than the C-terminal IDR, but the entire protein is involved in binding:

Construct	Construct boundaries (aa)	Kd (μ M)
TTP near FL	TTP (14-326)	5.5 \pm 0.6
TTP NTD+ZFD	TTP (14-171)	5.8 \pm 1.2
TTP ZFD+CTD	TTP (98-326)	9.6 \pm 1.8
TTP ZFD	TTP (98-171)	17.3 \pm 2.4

We note that the affinities here are low and there is a less than 2-fold difference between binding of full-length TTP, the NTD construct and the CTD construct. Further mapping revealed that multiple tryptophans distributed across the TTP IDRs contribute to binding and a W-containing peptide from the C-terminal IDR had the tightest binding (Kd ~9 μ M). Notably, isolated NOT9 dimerises due to lack of its binding partner NOT1.

We agree that the differences between our study and reference 24 are likely due to alternative experimental setups (full Ccr4-Not vs. NOT9 alone, and analysis of binding vs. deadenylation). We feel that it would be confusing to discuss these (small) differences in detail in the manuscript but we have adjusted our citation of this study: *The human RNA adaptor protein TTP (also known as ZFP36) selectively targets a set of pro-inflammatory mRNAs*³¹. *A conserved C-terminal SLiM¹² of TTP interacts with CNOT1 but additional regions in N- and C-terminal IDRs have also been implicated in binding*²⁴. We have also added another reference to this work in our discussion of the tryptophan binding pockets in the AlphaFold screen.

6. Fig. 5C: The phosphomimic mutations in TTP appear to perturb the activation of CCR4-NOT to a lesser extent than does phosphorylation of Puf3 (Fig. 4F). Is this due to lesser activation by wild-type TTP than by unphosphorylated Puf3 (leading to a smaller dynamic range)? Is it due to the limitations of phosphomimic mutations? Or does this reflect something deeper about the differences in the biological role of these two RNA adaptor proteins? The answers to these questions are worth noting in the manuscript text.

This is an excellent question. Phosphomimetic TTP (at 12 sites) reduces deadenylation by ~2.5-fold whereas phosphorylated Puf3 (at ~13 sites) reduces deadenylation by ~4.5-fold. The reasons for these differences could be due to the functional differences between phosphorylation vs. phosphomimetic mutations. Alternatively, it is possible that we have not hit the correct combination of modifications in our constructs.

Notably, the phosphorylated/phosphomimetic proteins activate deadenylation when compared to the RNA binding domain alone but they do so less than unmodified protein. This phosphorylation-mediated reduction in deadenylation activity (and thus the consequent change in mRNA half-life) could have substantial effects in cells. Indeed, phosphorylation is **known** to affect the activities of these RNA adaptors in cells, as cited in our manuscript. Our data provide a possible molecular mechanism for this. We have clarified this in the manuscript (new text in **bold**):

*...In summary, Puf3 is sequentially phosphorylated, which reduces (**but does not eliminate**) its interaction with Ccr4-Not and its ability to stimulate deadenylation activity...*

*...Interestingly, CCR4-NOT exhibits a graded response to TTP phosphorylation, similar to Puf3 phosphorylation (Fig. 5c and Extended Data Fig. 10d-e). **However, phosphomimetic mutations of TTP have a smaller impact on deadenylation activity compared with phosphorylation of Puf3.***

...

7. The caption for ED Fig 6 D is confusing. Please state the complex between Not9 and Not1 is depicted. '...AlphaFold2 prediction of *S. pombe* Not9 and Not1' suggest separate molecules might be displayed.

This is now **Extended Data Fig. 5**. We have updated the figure legend.

8. For ED Fig 7B in the PAE plots it looks like there may be trade-offs in accuracy of the predicted models Not9/Not1 with Puf3 containing one or two omega pockets are engaged. Do the authors have an interpretation of this observation?

We have removed these data from the manuscript and replaced them with our AlphaFold screens (new **Extended Data Fig. 5**). However In our new screens we still observe this phenomenon. We have two possible interrelated interpretations for this. Firstly our screens were performed with shorter fragments of the IDR which could lead to artificial placement of suboptimal motifs leading to lower pLDDT values. Secondly it could be that tandem binding in both pockets is creating a backbone strain that is also manifested in lower pLDDT values. Further biophysical examination of these particular interactions would be needed to disentangle these effects.

9. The authors should state why the MBP tag for Puf3 was not removed for biochemical studies.

The MBP tag acts as a solubility tag and prevents degradation of the IDR. We have also tested the effect of removing the MBP tag. Puf3 lacking the MBP tag stimulated deadenylation even more than MBP-Puf3. However, this results in a systematic error and the **relative** rates are similar. Thus, our conclusions are not affected. These data are now discussed in the manuscript, in the Methods and in **Extended Data Fig. 4a-c**.

Reviewer #3 (Remarks to the Author):

This work analyses molecular interactions between the adenylation complex Ccr4-Not and RNA adaptor proteins, such as Puf3. The interactions are characterized using biochemical assays (deadenylation assays), NMR spectroscopy and mass spectrometry-detected cross-links. The study aims at being an extensive biophysical and biochemical analysis of individual interactions and their functional significance. However, I am confused both by the presentation of the data, their analysis and the conclusions drawn from them. In details:

We thank the reviewer for their careful analysis of our manuscript. We have now collected new data, reanalysed data and improved the clarity. Overall, this has improved the clarity of our manuscript substantially.

1. In the absence of phosphorylation, activity assays are presented in Figure 1, Extended Data Figure 5 and 8. A relative quantification is done in Extended Data Figure 5 but not in the other Figures. For the mutants presented in ED Fig. 5, it is stated that they affect deadenylation efficiency much less than the deletion mutants in Fig. 1. How much less? Why are the data in Fig. 1 not quantified relatively to each other? Even for the $\Delta 258$ mutant, where the rate of deadenylation is probably too slow to be properly quantified, an upper limit could be provided.

We agree and have re-run the assays with more time points to quantify the relative activities. This is now integrated into **Fig. 1b** and the relevant Extended Data Figures. In addition, we now report $t_{1/2}$, instead of relative deadenylation activity so that the experiments in different figures are directly comparable.

2. The effect of the individual mutations of ED Fig. 5 is small and does not reach that of the $\Delta 258$ mutant even when combining all mutations. Nevertheless, it is argued that the effect of the mutations is additive. Can the authors specify what they mean by additive? Is this a quantitative or a qualitative statement? I am not sure that the word additive is appropriate, as I do not see that this “additivity” is in quantitative agreement with the data of ED Fig. 5. The additivity is nicely apparent in the selection of data given in Fig. 2. However, from ED Fig. 5C it appears that F65N has no additional effect when combined with L43N and W229N, while it has an effect by itself (ED Fig. 5D). On the other hand, an effect seems to exist for F65N in combination with L43N and W229N in Fig. 2. Similarly, Fig. 2 shows no effect for L43N in combination with W229N, while an effect is seen in ED Fig. 5C for L43N added to the combination W229N and F65N. Either the data of the two figures are inconsistent with each other or the effects are not additive.

The Reviewer is correct – ‘additive’ is not the correct term to describe the effects we see. Our data are consistent with multiple regions contributing to a multivalent interaction between the Puf3 IDR and Ccr4-Not. Their contribution is not necessarily additive. We have now modified the text accordingly throughout the manuscript. For example:

Discussion: Specifically, multiple regions within the IDRs of RNA adaptors contribute to the ability to recruit Ccr4-Not to specific transcripts and to stimulate its deadenylation activity, possibly in a cooperative manner.

3. The first half of the paper suggests that the affinity of the N-terminal disordered region of Puf3 for Ccr4-Not complex is correlated to the efficiency of deadenylation. While this is qualitatively supported by the data, it is not supported quantitatively. To obtain a quantitative support, the same mutants for which binding assays are presented in Figure 2 should also be tested in adenylation assays and vice-versa. For example, the affinities of Puf3109-258 and Puf3158-258 for Ccr4-Not differ by a factor of 10. How do the deadenylation activities of D109 and D158 compare with each other?

In our original submission, the deadenylation assays and binding studies were not performed with the same proteins – binding assays were performed with IDR segments that correlated with the NMR samples. Each type of analysis has some limitations:

- NMR samples are limited by size and therefore only contain the N-terminal portion of the IDR.
- Binding assays require high concentrations of homogenous proteins which we could not achieve some of the longer constructs. Thus, we originally chose to perform binding assays with the constructs used for NMR.

- Deadenylation assays require constructs with the PUM RNA binding domain. Therefore, to test the activity of some constructs, we needed to fuse the PUM domain to the upstream IDR, which alters the distance between them. Moreover, truncation mutants may result in a different orientation of bound RNA with respect to the active sites of Ccr4-Not, causing differences in activity not related to IDR-Ccr4-Not affinity but due to steric effects. The concentrations of proteins used in the assays (250 nM Puf3 construct, 50 nM Ccr4-Not) could result in limitations when using lower affinity peptides. Finally, given that Ccr4-Not deadenylates RNA even in the absence of Puf3, one would not expect a linear correlation between binding affinity and activity (a peptide with 100-fold less binding affinity would not have 100-fold less activity).

Nevertheless, we agree that it is useful to present binding studies and deadenylation assays with the same proteins. Thus, we have now performed a set of deadenylation assays with truncation mutants of the Puf3 IDR fused to the PUM domain (**Fig. 2b**). These new data show that there is generally a correlation between binding affinity and ability to stimulate deadenylation. There are some deviations but these are likely due to the limitations mentioned above and now discussed in the legend for **Extended Data Fig. 3**.

4. Why were affinities of Fig. 2 measured only with the N-terminal part? The disordered N-terminal region may have interaction in the context of the entire Puf3 protein that change its affinity for Ccr4-Not. The affinities of the N-terminal-only constructs should be compared with those of D109, D158 and D258 constructs.

The difficulties with the binding assays are discussed above. We have instead performed pull-down assays to systematically compare the relative affinities of different constructs which include the C-terminal PUM domain (**Fig. 2c, Extended Data Fig. 4d**). These data show that Puf3 residues 109-258 pull down a similar amount of Ccr4-Not as full-length Puf3, suggesting that the affinities are similar.

5. The NMR data should be analysed more carefully. A quick inspection of the spectra reveals that line shapes differ between apo and holo spectra. Instead of intensities, a more appropriate quantification would include FUDA fits of the peaks. No explanation is given about how the intensities were extracted. Was a fitting program used? How were spectra processed? All these factors influence intensity values. The effect in the region 10-80 are not continuous: is the assumption here that only individual amino acids contribute to the interaction with Ccr4-Not, while the rest are flexible? The data at different Ccr4-Not concentrations are not completely consistent. Any ideas why this is the case? In addition, the region between 210-260 seems to be less affected at higher concentrations of Ccr4-Not, which makes little sense. A proper fitting of the data, considering overlaps and excluding peaks that are too overlapped to be quantified, even with a proper line-fitting program, might shed light on these inconsistencies.

We agree that our original NMR data could have been more clearly presented. We had defined the interaction sites through a plot of relative peak intensity changes between Puf3 alone and Puf3 in the presence of Ccr4-Not (at a ratio of 1:10 Ccr4-Not:Puf3 IDR). These data were substantially exchange broadened and this likely contributed to noise in the intensity measurements and to the irregular pattern of intensities, especially in the region covering residues 10-80 (previous **Fig. 1d**).

Peak intensities were measured in POKY software and we have clarified this in the Methods section. This approach is in line with another recent study of larger IDRs (Naudi-Fabra et al., 2024, doi: 10.1038/s41467-024-50212-4). We attempted to measure peak volumes in POKY and NMR-FX. However due to the substantial peak overlap in large IDRs, fitting peak volumes was impractical.

To improve our work and strengthen the analysis, we have now modified our NMR approach. We collected data at 950 MHz (previously 800 MHz) and used lower ratios of Ccr4-Not (1:50, 1:100) to minimise line broadening effects. This reduces the non-continuous effect in residues 10-80 mentioned above, and the data across different Ccr4-Not concentrations are more consistent (new **Fig. 1c-d**). Because of the lower ratios of Ccr4-Not, we now observe only small changes in peak intensities and linewidths, either due to direct binding or conformational changes upon binding. To differentiate between these possibilities, we now also include ¹⁵N transverse relaxation data (R_2). This measures the dynamics of the conformational ensemble between the free and bound Puf3 IDR, allowing us to map the regions involved in binding in a more direct manner.

Our new data show increased R_2 rates around several distinct regions of the Puf3 IDR in the presence of Ccr4-Not. We have now added the following section to the text:

We also measured ^{15}N transverse relaxation rates (R_2) of Puf3-IDR¹⁻²⁵⁸ in the presence of Ccr4-Not, which can give an indication of which regions of Puf3 directly interact (Fig. 1e). We observed a small increase in R_2 across the whole IDR and more substantial increases in several distinct regions such as around residues 40-90 and 227-231 (Fig. 1e). This is consistent with a direct and dynamic interaction with Ccr4-Not, where the IDR is sampling both bound and unbound states.

The new data are much clearer and none of the conclusions have changed. With the addition of the new data, the CON experiments in the presence of Ccr4-Not do not add additional information and we have therefore removed these for clarity.

6. A discussion is needed about the exchange regime of the Puf3 N-terminal region with Ccr4-Not. Clearly, as the peaks disappear at sub-stoichiometric concentrations of Ccr4-Not, the exchange rate between bound and free forms of Puf3 is not slow. Have the authors attempted a more quantitative analysis of the line-broadening in dependence of the concentration? For example, the smaller line-broadening observed for the 10–80 region may indicate a lower affinity of this part of the protein for Ccr4-Not (i.e. a different exchange regime) or a different “rigidity” of this region when bound. Again, to prove the “additivity” argument, the contribution of the 10–80 region to the binding of Ccr4-Not should be investigated more thoroughly by quantitatively comparing the NMR data to the contribution of this region to affinity, as determined by other methods. Another possibility is that the 10–80 region does not have any direct interaction with Ccr4-Not but folds upon the 158–260 region (when this is interacting with Ccr4-Not) and stabilizes its interaction with Ccr4-Not. In this case, the presented conclusion about “multivalency” would be wrong.

The reviewer is correct that the exchange regime for the complexation between the Puf3 IDR and Ccr4-Not is fast. We exploited this fact to measure the R_2 rates in the presence of low (sub-stoichiometric) Ccr4-Not concentrations, as described above. Our transverse relaxation experiments are based on a CPMG effective B1 field of 1,000 Hz which would be suboptimal for measuring contributions from any slower (>1 ms) exchange regime. A detailed analysis of the contribution of exchange to the observed relaxation rates is beyond the scope of the present study so we cannot correlate increases in R_2 around distinct binding regions with their underlying affinity. Nevertheless, our new data show that the 1-108 construct plays a role in stimulating deadenylation (**Fig. 2b**) and binds very weakly in pull-down assays (**Fig. 2c**), and construct 1-158 binds Ccr4-Not with low affinity (**Extended Data Fig. 4e**). We could not quantify the binding affinities of all constructs, as we were not able to purify them in high enough concentrations.

Given that extensive regions of Puf3 and multiple distinct binding sites on Ccr4-Not are required for stimulation of deadenylation, our data is consistent with a multivalent mechanism.

7. The cross-link data do not seem to match well with the NMR data, but this fact is neither addressed nor discussed. Most of the cross-links seem to occur in the central region of Puf3, which is not consistently affected in NMR spectra. It is well known that cross-links may reveal also transient interactions and may overweight them with respect to more stable interactions. Is this what is happening here? Why do we not see any cross-links from the 210–260 region?

In this study, we employed the photoactivatable crosslinker SDA, which has been shown to significantly reduce the likelihood of overweighting transient interactions compared to commonly used NHS ester-based crosslinkers such as BS3 and DSSO (<https://doi.org/10.1101/2024.09.02.610668>).

Our analysis identified 54 self-links and 5 heteromeric crosslinks within the Puf3^{210–238} region, including two crosslinks to Not9. However, no crosslinks were detected between residues 238 and 260. In cross-linking mass spectrometry (CLMS), proteins are digested with trypsin to produce peptides optimal for mass spectrometry detection. The Puf3 sequence between residues 238 and 264 lacks trypsin cleavage sites, resulting in a minimum peptide length of 26 amino acids (**Fig. R2**), likely too long for efficient ionization and fragmentation after crosslinking to another peptide. (Individual peptides in crosslinked peptide pairs identified from the Ccr4-Not–Puf3 sample have an average length of 14 amino acids). In addition, given the large number of NHS-reactive groups in Puf3 residues 238-264, it is likely that this long peptide forms multiple covalent linkages, further reducing the likelihood of identifying cross linked peptides. We have now noted the following in the

legend for **Extended Data Fig. 5**: Due to poor coverage of trypsin cleavage sites, we could not detect peptides from all regions of the Puf3 IDR.

Figure R2: Correlation between the absence of crosslinks and trypsin cleavage site distribution in MBP-Puf3. MBP is highlighted in grey, Puf3^{210–260} (residues 601-651 in MBP-Puf3, 209-260 in Puf3 reference sequence) is highlighted in cyan. Proteins crosslinked to MBP-Puf3 are represented as circles. Crosslinks are shown as grey strokes (heteromeric-links) and arches (self-links). Trypsin cleavage sites (K and R residues) in MBP-Puf3 are indicated by red lines.

Because of these limitations, we have now performed systematic AlphaFold2 screens, scanning the Puf3 IDR in 50 residue windows, tiled every 3 residues, for binding to the known binding pockets on Ccr4-Not. These new data are presented in **Fig. 3b-c** and **Extended Data Fig. 5**. These data show that AlphaFold2 predicts that Puf3 binds in the peptide binding pockets of Ccr4-Not but it does not confidently identify exactly which regions of Puf3 bind. This raises the question of whether the peptide binding pockets each bind a single peptide or whether they can interact with multiple regions within the IDR. Both situations may occur, depending on the RNA adaptor. Overall, the AlphaFold screens provide important new insight into the interactions.

8. The observed cross-links are also not in agreement with the models presented in ED Fig. 6. In this figure only the region 210–260 of Puf3 interacts with Not9-Not1. Where is the region upstream of it that forms most cross-links?

As outlined above, the absence of crosslinks in the 238–264 region is most likely due to analytical limitations. The region upstream of 210-260 crosslinks to Not9 and to other regions of Puf3, indicating spatial proximity. These data also correlate with our new NMR relaxation data showing that residues around 200 are more flexible (low R_2 rates in the presence of Ccr4-Not **Fig. 1e**).

To further address this discrepancy we have now performed a new systematic AlphaFold interaction screen of the IDR with Ccr4-Not binding regions which includes the upstream region, as described above (see Methods section, **Fig. 3b-c** and **Extended Data Fig. 5**).

9. In ED Fig. 6 the prediction quality seems to be higher for the individual residues that interact with the aromatic pocket(s) than for the residues that interact with the peptide binding pocket, whose prediction score is rather poor. I am not sure what to make of it in terms of the overall trustability of these models. Probably quite poor overall.

In our new screen the highest confidence interactions are centred around tryptophans. We agree that AlphaFold does not perform well in predicting these weak interactions, underscoring the importance of our experimental data.

10. The phosphorylation bit is valuable, but again I do not understand the experimental design. Why were NMR experiments measuring relaxation performed only for Puf3109-258, if we know that the

residues 10–80 also bind Ccr4-Not? In the paper, this is justified by the need to reduce heterogeneity, but I do not understand which heterogeneity is meant here. At least, it should be checked that relaxation parameters of the 109–258 stretch w/o phosphorylation do not change between the 1–258 and 109–258 constructs.

We purified phosphorylated IDR away from kinases. We also attempted to separate different phosphostates, but the IDR is not uniformly phosphorylated. This mixture of different phosphorylated species results in heterogeneity in the NMR spectra. The shorter construct minimised the complications due to this heterogeneity.

Phosphorylation leads to local changes in the backbone dynamics of the IDR. We have analysed the relaxation of 1-258 and 109-258 constructs and find that there is no difference in the local behaviour between these two constructs (**Extended Data Fig. 2b**). Thus, it is very likely that our conclusions based on analysis of the shorter phosphorylated construct also apply to these sites when embedded in a longer construct.

11. The authors seem to assume that individual disordered stretches in the context of a long, disordered contrasts do not interact with each other. This is evident from the experimental design (see previous point) and from the last paragraph (before Discussion) supposedly presenting proofs of a general multivalency recognition mechanism. This assumption has been demonstrated wrong in many papers and is a rather simplistic way to view disordered proteins. These days we know that disordered–disordered interactions are functional. When they observe an increasing functional impairment upon deletion of longer IDR stretches, the authors conclude that this data indicates the presence of multivalent interactions. This is not necessarily the case. Disordered regions have internal interactions and residual structures and are not equal to beads on a chain. Individual disordered stretches of a long, disordered domain cannot be treated as independent from each other. Thus, the experimental data presented in the last paragraph do not support the conclusion of a “universal” multivalency mechanism (see also point 6). Multivalency can be at play here, but the data do not demonstrate it robustly enough and further experiments/analysis are needed.

We do not exclude that the IDR may interact with itself – this could be possible. However, and importantly, we show that there is a requirement for long stretches of the IDR for function (deadenylation activity), and we also show that **multiple peptide binding pockets on Ccr4-Not** are required for the interaction. Together, the NMR, binding data and deadenylation assays provide strong support for a multivalent interaction. We could not find the word “universal” in the main text. We have now altered the text appropriately to emphasise that our data are **consistent with a multivalent binding mechanism**:

End of Results section: *Thus, multiple RNA adaptors likely bind to multiple sites on Ccr4-Not using extended regions of their IDRs, that can be regulated in a tunable manner to control mRNA deadenylation.*

Discussion: *In this study, we demonstrate that the yeast RNA adaptor protein Puf3 interacts with the Ccr4-Not deadenylase complex through a multivalent mechanism that can be regulated in a rheostat-like manner (Fig. 5d). Our data are also consistent with human RNA binding proteins (PUM1 and TTP) using multiple interaction sites for Ccr4-Not binding and regulation. Specifically, multiple regions within the IDRs of RNA adaptors contribute to the ability to recruit Ccr4-Not to specific transcripts and to stimulate its deadenylation activity, possibly in a cooperative manner.*

12. To ED Figure 15, in the example of TTP, the deletion of the N-term and C-term are all but additive! There is cooperativity between the N-term and the C-term, which goes beyond multivalency.

We agree that we have not demonstrated that the N-terminus and C-terminus of TTP act in an additive manner.

Other points:

In the introduction, it is confusing to use many different names for the same protein. If this is done, then it should be explicitly stated to which organism each name applies. The way it stands is jargon for experts.

Many abbreviations are not spelled out.

We agree that the nomenclature is confusing. Many people have different preferences for subunit names. We prefer to list all of the orthologous subunit names in the main text and have added the species. These are the protein names and not necessarily useful abbreviations.

In conclusion, although the topic of the paper is of interest, the biochemical and biophysical analysis lacks the consistency and rigor that I would expect for a manuscript in NSMB. The concept of multivalency is trivial if not substantiated by a quantitative analysis. This work lacks a rigorous quantification of the contributions to binding of the different regions of Puf3, the multiple, interdisciplinary assays are not applied to a consistent set of constructs, the NMR data are superficially analysed, the cross-link data are inconsistent with all other data and there is no information of which part of Puf3 interacts with which part of Ccr4-Not. In addition, the authors do not consider that a long, disordered region of 258 residues may not be a series of bids on a chain but may entertain disorder-to-disorder interactions. These interactions underline the necessity of using the full-length construct or at least of running the appropriate controls. In addition, data interpretation need to be revised.

We have addressed all of the Reviewers concerns with new data, more quantitations, better explanations and more clarity in the manuscript.

Point-by-point response to Reviewer comments

Our responses to the Reviewers are in blue.

Reviewer #1:

Remarks to the Author:

The authors have addressed all my concerns in this revision and substantially strengthened their manuscript. This is an important study for the RNA decay field and for IRD research, and I am looking forward to seeing it published as soon as possible.

Georg Stoecklin

Reviewer #2:

Remarks to the Author:

The authors have addressed my questions and comments in their revised manuscript. It's an outstanding study that uses a wide range of techniques to biochemically characterize an extremely challenging system. The work goes beyond just the area of deadenylation/mRNA decay, given its quantitative dissection of the multivalent interaction and the "rheostat" model of tuning via phosphorylation.

Minor suggestion: In the legend for Fig 4e, the authors should indicate that red and black bars correspond to phosphorylated and unphosphorylated forms of Puf3, respectively.

We have made this change.

Additionally, the authors have addressed the comments and questions of Reviewer 3. Most notably, they have clarified the rationale for the constructs studied by NMR, fluorescence anisotropy and deadenylation assays and have added additional protein interaction (pull-down) experiments to address the correlation between binding and the ability of the Puf3 IDR to stimulate deadenylation activity. Additionally, they have fortified their NMR analysis with measurements of ¹⁵N R2 relaxation rates for the IDR of Puf3 to monitor changes in backbone mobility as a function of binding to Ccr4/Not and phosphorylation. Lastly, they have changed their interpretation of 'additivity' of IDR regions, taking into account comments on cooperative interactions as evidenced by mutational data.

Overall, this manuscript is now suitable for publication in NSMB.

Reviewer #4:

Remarks to the Author:

The authors have addressed the criticisms raised by the reviewers in a constructive manner. In particular, the new "quantifications" and the analysis of the correlation between deadenylase activity and molecular interactions strengthen the conclusions and add depth to the study. The manuscript in its current form represents a significant contribution to the field and is suitable for publication.

There is one minor inconsistency in the response to point 3, where Extended Data 3 is referenced, though it appears the correct figure is Extended Data 4 ("There are some deviations but these are likely due to the limitations mentioned above and now discussed in the legend for Extended Data Fig. 3")

We apologise for the error in the previous response letter.